# Molecular structures of the human Slo1 K$^+$ channel in complex with β4

Xiao Tao, Roderick MacKinnon*

Laboratory of Molecular Neurobiology and Biophysics, The Rockefeller University, Howard Hughes Medical Institute, New York, United States

**Abstract** Slo1 is a Ca$^{2+}$- and voltage-activated K$^+$ channel that underlies skeletal and smooth muscle contraction, audition, hormone secretion and neurotransmitter release. In mammals, Slo1 is regulated by auxiliary proteins that confer tissue-specific gating and pharmacological properties. This study presents cryo-EM structures of Slo1 in complex with the auxiliary protein, β4. Four β4, each containing two transmembrane helices, encircle Slo1, contacting it through helical interactions inside the membrane. On the extracellular side, β4 forms a tetrameric crown over the pore. Structures with high and low Ca$^{2+}$ concentrations show that identical gating conformations occur in the absence and presence of β4, implying that β4 serves to modulate the relative stabilities of 'pre-existing' conformations rather than creating new ones. The effects of β4 on scorpion toxin inhibition kinetics are explained by the crown, which constrains access but does not prevent binding.

## Introduction

The Slo1 channel, also known as BK or MaxiK, distinguishes itself from other K$^+$ channels by its unusually large single-channel conductance (~10–20 times greater than most other K$^+$ channels) and dual regulation by intracellular Ca$^{2+}$ and membrane voltage (*Marty, 1981*; *Pallotta et al., 1981*; *Barrett et al., 1982*; *Latorre et al., 1982*; *Latorre and Miller, 1983*; *Elkins et al., 1986*; *Atkinson et al., 1991*; *Adelman et al., 1992*; *Butler et al., 1993*; *Pallanck and Ganetzky, 1994*; *Magleby, 2003*). Because both Ca$^{2+}$ and membrane voltage gate Slo1, it serves as a hub in numerous physiological processes that couple membrane excitability to Ca$^{2+}$ signaling events such as muscle contraction, audition, hormone secretion, and neurotransmitter release (*Contreras et al., 2013*; *Latorre et al., 2017*). Deficiencies in the Slo1 channel have been linked to a spectrum of diseases including hypertension, urinary incontinence secondary to overactive bladder (OAB), epilepsy, mental retardation, and autism (*Contreras et al., 2013*; *Latorre et al., 2017*).

Slo1 functions as a tetramer of the pore-forming α subunit, which is encoded by a single gene, *Slowpoke* (*KCNMA1*). Previously, we determined the atomic structures of aplysia Slo1(acSlo1) in the absence and presence of Ca$^{2+}$ (*Hite et al., 2017*; *Tao et al., 2017*). These structures provided an explanation for the apparent paradox of an unusually high conductance and exquisite K$^+$ selectivity, showed how the Ca$^{2+}$ sensing mechanism can work, and showed how the Ca$^{2+}$ and voltage sensors are in contact with each other, allowing for the possibility that the two sensing mechanisms could be directly coupled (*Hite et al., 2017*; *Tao et al., 2017*).

In mammals, Slo1 channels usually consist of α plus auxiliary subunits. The α subunits are similar to those of invertebrate Slo1 channels, but the auxiliary subunits represent a novel feature. Auxiliary subunits give rise to functional diversity and to tissue-specificity amongst the otherwise ubiquitously expressed Slo1 channel α subunit. There are two distinct families of auxiliary subunits discovered so far, termed β (2-transmembrane, 2-TM) and γ (single-TM) (*Garcia-Calvo et al., 1994*; *Knaus et al., 1994b*; *Knaus et al., 1994c*; *Behrens et al., 2000*; *Brenner et al., 2000*; *Meera et al., 2000*; *Uebele et al., 2000*; *Weiger et al., 2000*; *Xia et al., 2000*; *Braun, 2010*; *Yan and Aldrich, 2010*;

*For correspondence:
mackinn@mail.rockefeller.edu

Competing interests: The authors declare that no competing interests exist.

*Yan and Aldrich, 2012*; *Zhang and Yan, 2014*). These auxiliary subunits diversify the function of Slo1 to the greatest extent compared to other modifications such as alternative splicing, phosphorylation etc. They dramatically modify nearly all aspects of Slo1's biophysical properties (including the activation and deactivation kinetics and $Ca^{2+}$ sensitivity) as well as pharmacological characteristics (such as the channel's sensitivity to natural toxins) (*McManus et al., 1995*; *Dworetzky et al., 1996*; *Chang et al., 1997*; *Orio et al., 2002*; *Lippiat et al., 2003*; *Ha et al., 2004*; *Brenner et al., 2005*; *Wang et al., 2006*; *Savalli et al., 2007*; *Martin et al., 2008*; *Braun, 2010*; *Wu and Marx, 2010*; *Yan and Aldrich, 2010*; *Yan and Aldrich, 2012*; *Torres et al., 2014*; *Zhang and Yan, 2014*; *Latorre et al., 2017*). There are four members identified for each family (β1-β4 and γ1-γ4) and members within the same family modify channel function to very different extents. These auxiliary subunits generally do not exist in lower animals. Therefore, regulation of Slo1 by auxiliary subunits appears to be a key mechanism of functional tuning to fulfill different physiological roles in various tissues and cell types of higher animals.

How these single or 2-TM small transmembrane proteins physically associate with and modulate the large pore-forming α subunit's function has been studied by many scientists since the discovery of the first member - β1 in 1994 (*Garcia-Calvo et al., 1994*; *Knaus et al., 1994b*; *Knaus et al., 1994c*). So far, no structure of a β or γ subunit alone or in complex with the α subunit has been determined to help us understand how these proteins work. In this study we present the structures of a mammalian (human) Slo1 channel consisting of the α subunit alone and in complex with the brain-enriched β4 subunit. In each case (presence and absence of β4 subunit) we have determined the structures of $Ca^{2+}$-activated (open) and $Ca^{2+}$-depleted (closed) conformations. We also interrogate the influence of structure-guided mutations of β4 using an electrophysiological assay.

## Results

### Structure determination of the human Slo1 channel in 4 states

To obtain biochemically stable protein samples suitable for structural studies we modified the gene encoding the Slo1 α subunit by removing the C-terminal 57 amino acids (unstructured in the crystal structure of the human Slo1 cytoplasmic domain) (*Yuan et al., 2010*). This modification neither affected the function of the Slo1 channel α subunit alone nor the functional influence β subunits have on Slo1 (*Figure 1—figure supplement 1*). Co-expression with either the β1 or β4 subunit slowed the activation and deactivation kinetics of the truncated Slo1 channel and modified its apparent $Ca^{2+}$-sensitivity, as reported for the full length Slo1 channel (*Figure 1—figure supplement 1*) (*McManus et al., 1995*; *Dworetzky et al., 1996*; *Lippiat et al., 2003*; *Ha et al., 2004*). The truncated Slo1 channel fused with a C-terminal GFP was either expressed alone or co-expressed with the β4 subunit in HEK293S GnTI⁻ cells using the BacMam method (*Goehring et al., 2014*) and purified in the presence of Digitonin and a mixture of phospholipids using a GFP nanobody-affinity column followed by size-exclusion chromatography (*Fridy et al., 2014*). The final protein sample of Slo1 co-expressed with the β4 subunit contained both the α subunit and the β4 subunit confirmed by SDS-PAGE and mass spectrometry. Structures of human Slo1 in four distinct states were determined: $Ca^{2+}$-free and $Ca^{2+}$-bound α subunit alone as well as $Ca^{2+}$-free and $Ca^{2+}$-bound α-β4 complex at resolutions of 4.0 Å, 3.8 Å, 3.5 Å and 3.2 Å, respectively (*Figure 1A*, *Figure 1—figure supplements 2–4* and *Table 1*). Details of the structure determinations are given in Materials and methods.

The final reconstruction of the $Ca^{2+}$-bound α-β4 complex is of highest quality compared to the others, with the majority of regions well resolved for model building (*Figure 1—figure supplements 2* and *3*). The resolution of the density corresponding to the β4 extracellular region was worse than the TM regions (*Figure 1—figure supplements 2F* and *3*). Still we were able to build an essentially complete model of β4 de novo using a subclass from focused-classification (see Materials and methods). Registration of the β4 subunit sequence was confirmed by recognizable large sidechains and the presence of four disulfide bonds as well as sugars attached to the two sequence-predicted and mass-spectrometry-confirmed N-glycosylation sites. The final model has good geometry and contains amino acids 16–54, 91–569, 577–615, 681–833 and 871–1056 of the α subunit and amino acids 7–205 of the β4 subunit (*Table 1*).

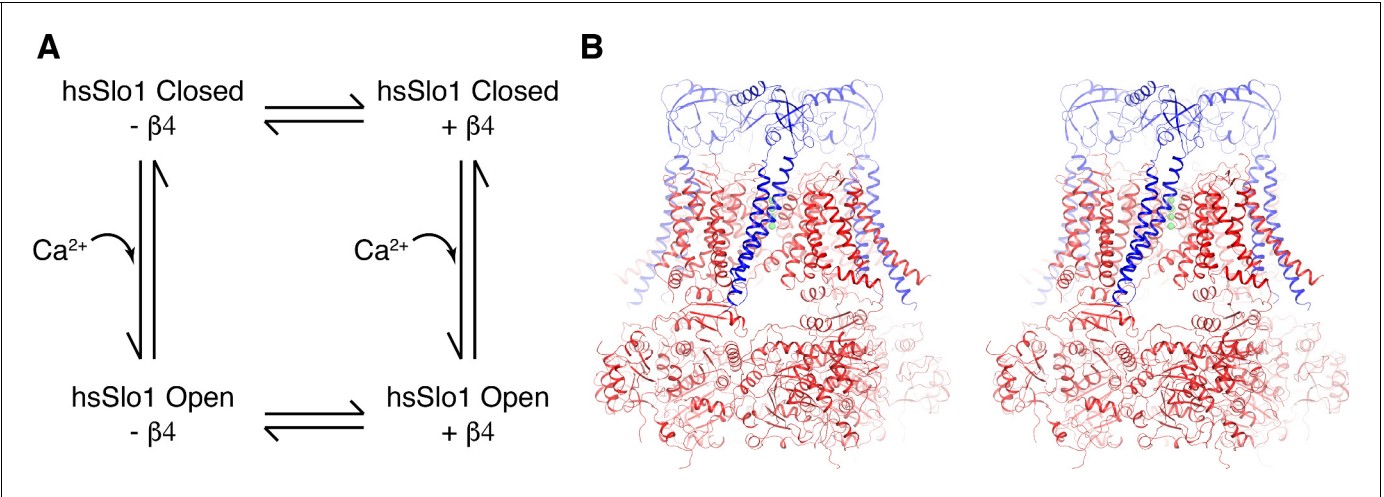

**Figure 1.** Overall structure of the open human Slo1 channel in complex with the β4 subunit. (**A**) Determination of atomic structures of human Slo1 in four various states. (**B**) Overall structure of the human Slo1-β4 channel complex in the presence of 10 mM $Ca^{2+}$ in stereo, viewed parallel to the membrane. The Slo1 channel and β4 subunits are shown in ribbon representation, and colored red and blue respectively. Green spheres represent the $K^+$ ions in the selectivity filter.

The online version of this article includes the following figure supplement(s) for figure 1:

**Figure supplement 1.** Electrophysiological studies of truncated hsSlo1 alone, co-expressed with β1 or β4 in HEK293T cells.

**Figure supplement 2.** Structure determination of the open Human Slo1 channel in complex with β4 using Cryo-EM.

**Figure supplement 3.** Individual EM densities of the open human Slo1 channel in complex with β4.

**Figure supplement 4.** Structure determination of the closed human Slo1 channel in complex with β4, the open human Slo1 channel, and the closed human Slo1 channel using Cryo-EM.

The above atomic model of the $Ca^{2+}$-bound α-β4 complex was used as a starting model for the other three conformations, followed by multiple rounds of manual rebuilding in Coot and real-space refinement with Phenix (*Emsley et al., 2010*; *Afonine et al., 2018*). The final models all have good geometry (*Table 1*). Due to its highest resolution, the atomic model of the $Ca^{2+}$-bound α-β4 complex is used for most of the structural description and analysis in this manuscript.

## Quaternary structure of the open human Slo1 α-β4 channel complex

The human Slo1 α subunit tetramer is organized similarly to acSlo1 (*Figure 1B*) (*Tao et al., 2017*). Four of the β4 subunits bind to the human Slo1 tetramer, extending the channel extracellularly by about 40 Å when viewed from the side (*Figure 1B*). The complex has dimensions of approximately 150×150×150 Å. Consistent with disulfide crosslinking data in the literature, β4 subunits are located between voltage sensor domains (VSDs) and each β4 subunit contacts two neighboring VSDs simultaneously (*Figure 1B*) (*Liu et al., 2008*; *Wu et al., 2009*; *Liu et al., 2010*; *Wu et al., 2013*). The 120-amino acid linker between the two TMs (TM1 and TM2) of β4 forms a well-ordered structure on the extracellular side (termed 'EC domain' throughout the manuscript). Four EC domains form a 'crown' on top of the Slo1 channel (*Figure 1B*).

The protein sample contained high concentrations of $Ca^{2+}$ and $Mg^{2+}$ (10 mM each). As expected, the channel adopts an open conformation and both $Ca^{2+}$ binding sites (the Ca-RCK1 site and the Ca-bowl site) in the gating ring and the $Mg^{2+}$ binding site at the interface between RCK1 and the VSD are occupied, as observed in acSlo1 under the same buffer conditions (*Tao et al., 2017*).

## Structure of the β4 subunit

*Figure 2A* shows a ribbon representation of the β4 subunit monomer in stereo. As predicted from the primary sequence, the β4 subunit contains two long transmembrane helices, TM1 and TM2 (*Figure 2A*). Density for the N-terminal six amino acids as well as the C-terminal five amino acids (residues 206–210) were not visible in the open Slo1-β4 complex, indicating their structural flexibility. Amino acids 7–11 form a short loop (termed 'N-loop') preceding TM1 (*Figure 2A*). TM1 kinks near

**Table 1.** Structure refinement and validation, related to *Figure 1* and *Figure 5*.

| | hsSlo1 + β4 Open | hsSlo1 + β4 Closed | hsSlo1 Open | hsSlo1 Closed |
|---|---|---|---|---|
| **Data acquisition** | | | | |
| Microscope/Camera | | Titan Krios/Gatan K2 Summit | | |
| Voltage (kV) | | 300 | | |
| Defocus range (μM) | 0.7 to 2.0 | 0.7 to 2.0 | 0.8 to 2.4 | 0.8 to 2.4 |
| Pixel size (Å) | 1.04 | 1.3 | 1.3 | 1.3 |
| Total electron dose (e⁻/Å²) | 74 | 89 | 89 | 89 |
| Exposure time (s) | 10 | 15 | 15 | 15 |
| **Reconstruction** | | | | |
| Particle number | 117,791 | 42,842 | 28,073 | 53,961 |
| Resolution (unmasked, Å) | 3.7 | 4.0 | 4.2 | 4.4 |
| Resolution (masked, Å) | 3.2 | 3.5 | 3.8 | 4.0 |
| **RMS deviation** | | | | |
| Bond length (Å) | 0.007 | 0.007 | 0.007 | 0.01 |
| Bond angle (°) | 0.801 | 0.946 | 0.950 | 0.94 |
| **Ramachandran plot** | | | | |
| Favored (%) | 95.55 | 91.70 | 91.72 | 90.43 |
| Allowed (%) | 4.45 | 8.30 | 8.16 | 9.46 |
| Outliers (%) | 0.00 | 0.00 | 0.12 | 0.11 |
| **MolProbity** | | | | |
| Clash score | 5.83 | 6.07 | 8.56 | 5.36 |
| Rotamer outliers (%) | 0.21 | 0.30 | 1.56 | 0.63 |

the extracellular membrane interface and extends further into the extracellular space about four additional helical turns while TM2 extends beyond the intracellular membrane interface into the cytoplasm (*Figure 2A*). TM1 and TM2 within the β4 monomer form an extensive interface with mostly hydrophobic interactions (*Figure 2B*). The extended C-terminal part of TM2 also interacts with the N-loop through a few hydrogen bonds (*Figure 2B*).

The well-ordered EC domain, which connects TM1 and TM2, contains mostly β strands (βA-βE, βH-βI), two short α helices (αF-αG) and loops in between (*Figure 2A and C* and *Figure 2—figure supplement 1A*). The structure is held together by four disulfide bonds (*Figure 2C*). The β4 EC domain also contains two predicted N-glycosylation sites, with one of them conserved among all the β subunits (*Figure 2C* and *Figure 2—figure supplement 1A*). Glycosylation at these two predicted sites were confirmed with tandem ms/ms and extra density near both sites was observed in the final map, most likely representing the sugars. Glycosylation was shown to regulate the sensitivity of Slo1-β4 to iberiotoxin and also modify other biophysical properties of Slo1-β1 (*Jin et al., 2002*; *Hagen and Sanders, 2006*). In the open Slo1-β4 model, these two glycosylation sites are located at the outer periphery of the EC domain, far from the β4/β4 or α/β4 interfaces (*Figure 2C*). Therefore, the structure does not obviously explain the functional effects of these modifications. We note that proteins in this study were expressed using a glycosylation-deficient strain that produces smaller sugar groups. We cannot rule out the possibility that these sugars under physiological conditions might form carbohydrate-carbohydrate or carbohydrate-protein interactions not observed in the current structural model. The secondary structural elements as well as the four disulfide bonds match a recently published NMR structure of the extracellular loop of human β4 (*Wang et al., 2018*). However, our Slo1-β4 complex structure exhibits a completely different tertiary structure of the EC domain than that proposed on the basis of NMR experiments. Based on a hypothetical Slo1-β4 model constructed from NMR titration and the β4 EC solution structure, N123 on β4 was suggested to be critical for regulating Slo1 gating through interactions with E264 on the turret of the α subunit upon Ca²⁺ binding (*Wang et al., 2018*). The distance between sidechains of these two residues was

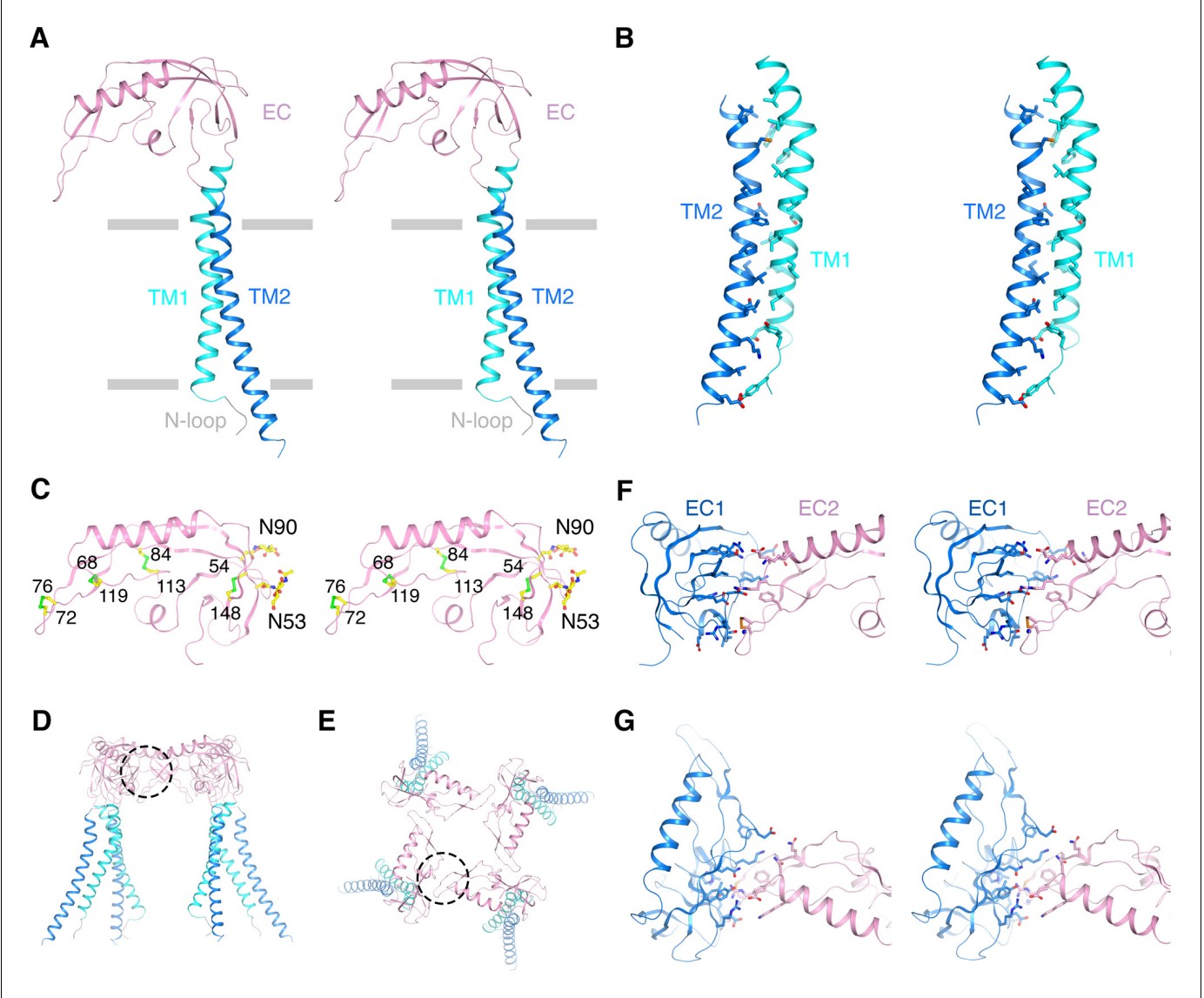

**Figure 2.** Architecture of the β4 subunit. (A) Stereo view of the β4 subunit monomer in ribbon representation with the extracellular side up. The N-terminal loop ('N-loop'), TM1, extracellular domain ('EC') and TM2 are discretely colored gray, cyan, pink and blue. The gray bars delimit the membrane boundaries. (B) TM1 and TM2 within one β4 subunit interact extensively with each other. TM1 and TM2 are shown as ribbons in stereo view. Residues involved in the interactions are shown in sticks and colored according to atom type. (C) The EC domain of β4 subunit in stereo, viewed parallel to the membrane. The protein is shown as ribbons and colored as in panel (A). Four disulfide bonds and 2 N-glycosylation sugar groups are shown as sticks and colored according to atom type. (D, E) The β4 subunit tetramer in ribbon representation viewed parallel to the membrane (D) or from the extracellular side (E). Interfaces between two neighboring EC domains are highlighted by dotted circles. (F, G) The interface between the EC domain of two neighboring β4 subunits viewed parallel to the membrane (F) or from the extracellular side (G). The two EC domains are colored blue and pink. Sidechains of residues at the interface are shown as sticks and colored according to atom type.

The online version of this article includes the following figure supplement(s) for figure 2:

**Figure supplement 1.** Sequence alignments.

hypothesized to shift from 10.6 Å to 5.5 Å when the channel opens, allowing them to interact. However, these two residues are more than 27 Å apart in our open Slo1-β4 complex structure and remained essentially static in the closed Slo1-β4 complex structure, making any direct interactions very unlikely.

No structural homologs of the EC domain have been reported. A search of the protein data bank using Dali revealed only proteins sharing part of the structure with the top hits being mostly

nucleotide binding proteins (such as exosome complex component RRP45 and translation initiation factor 2 γ subunit), which contain some of the β strands, and certain toxins (such as pertussis toxin and subtilase cytotoxin), which contain much shorter helices and loops. The physiological importance of this EC domain must await further study (see Discussion).

The stoichiometry of α and β subunits was assumed generally to be 1:1. Evidence of sub 1:1 stoichiometry emerged from analysis of co-expression in oocytes (*Wang et al., 2002*) as well as in native tissues (*Solaro et al., 1995*; *Ding et al., 1998*; *Kuntamallappanavar et al., 2017*). It was proposed that a Slo1 channel can contain zero to four β subunits, with each β subunit incrementally influencing channel gating properties (*Wang et al., 2002*). In the Slo1-β4 complex structure, β4 binds to the α subunit with 1:1 subunit stoichiometry (i.e. four β4 subunits per tetramer of α subunits) (*Figures 1B* and *2D–E*). The four β4 subunits form extensive interfaces with neighboring EC domains, encompassing a buried area of ~907 $\text{Å}^2$/monomer (*Figure 2D–G*). The protein sample used for EM studies was produced by overexpressing β4, thus we cannot rule out the possibility that a sub 1:1 β:α complex could exist under physiological conditions. However, the structure would predict a more stable conformation upon tetramer formation because the EC domain, which interacts extensively with itself, makes only minimal contact with the α subunit (*Figures 1B* and *2D–G*).

## Chemical nature of interfaces between β4 and α

Many biochemical and molecular biological studies have analyzed interactions between Slo1 and β subunits using, for example, disulfide crosslinking and TOXCAT assays (*Liu et al., 2008*; *Wu et al., 2009*; *Liu et al., 2010*; *Morera et al., 2012*; *Wu et al., 2013*). TM1 was predicted to be in the vicinity of S1 and S2 on the VSD (*Liu et al., 2008*; *Wu et al., 2009*; *Liu et al., 2010*; *Wu et al., 2013*) and direct interactions between TM1 of β2 and S1 of the VSD were found using the TOXCAT assay (*Morera et al., 2012*). In our Slo1-β4 structure, β4 associates with the α subunit almost exclusively through the two TM helices (*Figure 3A*). The TM1 segment near the outer leaflet of the membrane contacts the transmembrane domain (TMD) from one α subunit ('α1') through hydrophobic interactions with sidechains from S1 and the pore helix (PH), while at the same time interacting with sidechains from S6 and the turret of a contiguous α subunit ('α2') (*Figure 3A–B*). Notably, lipid molecules also contribute to these interfaces (*Figure 3A–B*). The TM1 segment near the inner leaflet acyl chain region mostly contacts S3 of α2 and several ordered lipid molecules through hydrophobic interactions (*Figure 3C*). The bottom of TM1, near the intracellular membrane interface, makes contacts with multiple regions of the α subunit including the S2-S3 linker and S0 from α2, the S6-RCK1 linker from α1 and phospholipid headgroups through hydrogen bonds (*Figure 3C*). With the same TOXCAT assay, no direct association with the α subunit was detected for TM2, the EC domain, or the C-terminus of β and it was thus suggested that there are no strong interactions between TM2 and S0 despite the nearness predicted from the disulfide crosslinking experiments (*Liu et al., 2008*; *Wu et al., 2009*; *Liu et al., 2010*; *Morera et al., 2012*; *Wu et al., 2013*). Our structure shows that the TM2 outer leaflet segment only interacts with α2. Furthermore, this interface is purely mediated by lipid molecules forming a hydrophobic core with TM2 sidechains (*Figure 3A–B*). The TM2 inner leaflet segment forms an extensive protein:protein interface with S0 and S3 from α2 (*Figure 3A and C*). The residues of β4 directly involved in the α/β association turn out to be fairly conserved among β1-β4 (*Figure 2—figure supplement 1A*). Therefore, it would seem likely that β1-β4 subunits all bind to Slo1 in a similar fashion. Furthermore, Slo1 residues involved in these specific interactions appear to be conserved between Slo1 and Slo3, which would explain why all members of the β subunit family also physically associate with Slo3 (*Figure 2—figure supplement 1B*) (*Yang et al., 2009*). We emphasize the general observation that lipid molecules make significant contributions to the α/β interfaces at both the inner and outer membrane leaflets, suggesting they are an integral part of the Slo1 α-β4 channel complex (*Figure 3*).

In the open α-β4 model, the N-loop preceding TM1 is located adjacent to the S6-RCK1 linker and the αC helix of the RCK1 N-lobe, making short and long-range interactions between these regions likely. T11 of β4 is less than 5 Å away from D173 in the S2-S3 loop of the α subunit; the E12 sidechain of β4 is about 5.5 Å away from R329 in the S6-RCK1 linker and is also in the vicinity of W176 on S3 and H409 on the αC helix.

Wang et. al. suggested that the EC domain of β1 might interact with the extracellular side of the Slo1 VSD, as mutations in that region caused appreciable changes in Slo1 gating (*Gruslova et al., 2012*). The Slo1-β4 complex structure shows essentially no direct association between the β4 EC

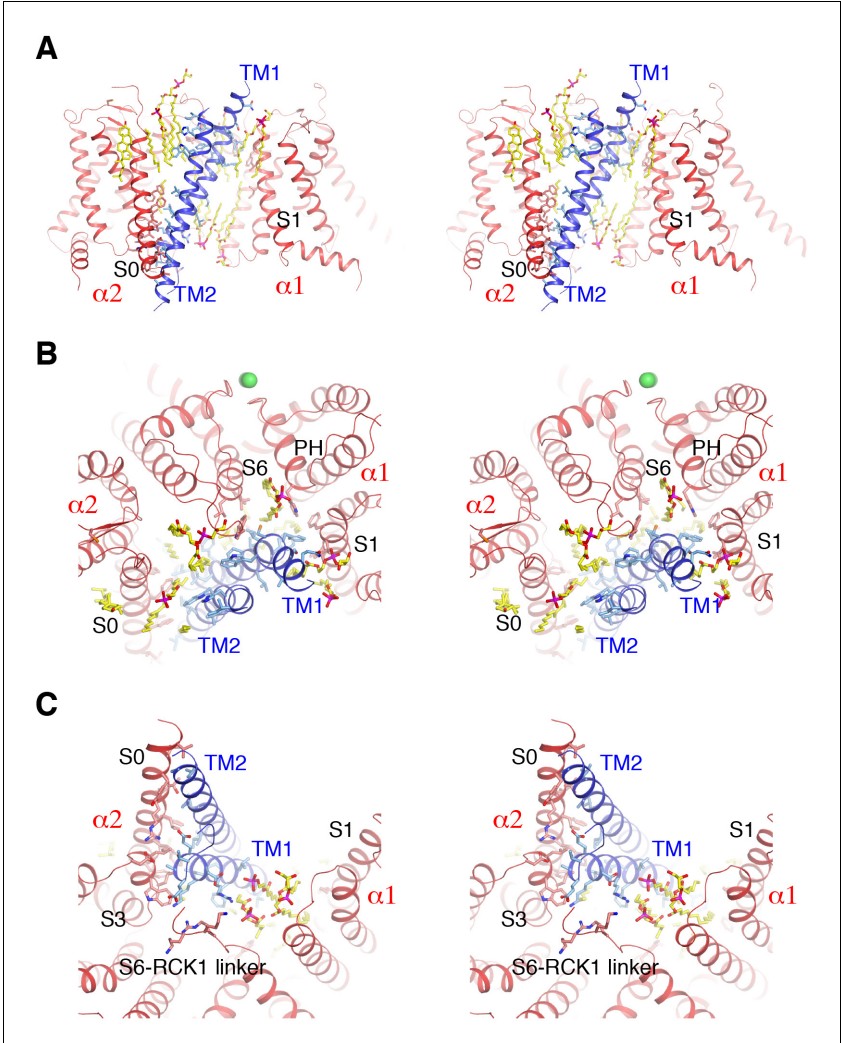

**Figure 3.** Detailed interactions between Slo1 and β4. (**A**) Extensive interactions between TM1, TM2 of β4 subunit and transmembrane domains of two contiguous Slo1 subunits (α1 and α2) in stereo. β4 and Slo1 are shown as ribbons and colored blue and red, respectively. Sidechains as well as lipids at the interfaces are shown as sticks and colored according to atom type. (**B**) β4 and Slo1 interface near the outer leaflet of the membrane in stereo, viewed from the extracellular side. Proteins and lipids are represented and colored as in panel (**A**). Green spheres represent the $K^+$ ions in the selectivity filter. (**C**) β4 and Slo1 interface near the inner leaflet of the membrane in stereo, viewed from the intracellular side. Proteins and lipids are represented and colored as in panel (**A**).

domain and the α subunit. The functional effects they observed from the mutations could be due to indirect effects.

## Influence of β4 N-terminus on Slo1 gating

β1 and β4 subunits slow the kinetics of Slo1 activation and deactivation in response to voltage steps (*Figure 1—figure supplement 1*; *Figure 4A*). We sought to identify regions of β4 that mediate these rate changes. Guided by the structure, we divided β4 into 10 regions depicted in *Figure 4B* and described in detail in Materials and methods. Mutants were made by replacing one or more of these 10 regions of β4 mainly with the equivalent sequence from β1 (*Figure 4B* and *Table 2*). The β1 sequence – instead of alanine or other substitutions – was used, reasoning this would more likely achieve expression and assembly of a functional complex. Co-expression was carried out in Xenopus oocytes and currents were recorded under two-electrode voltage clamp (TEVC), as shown (*Figure 4*).

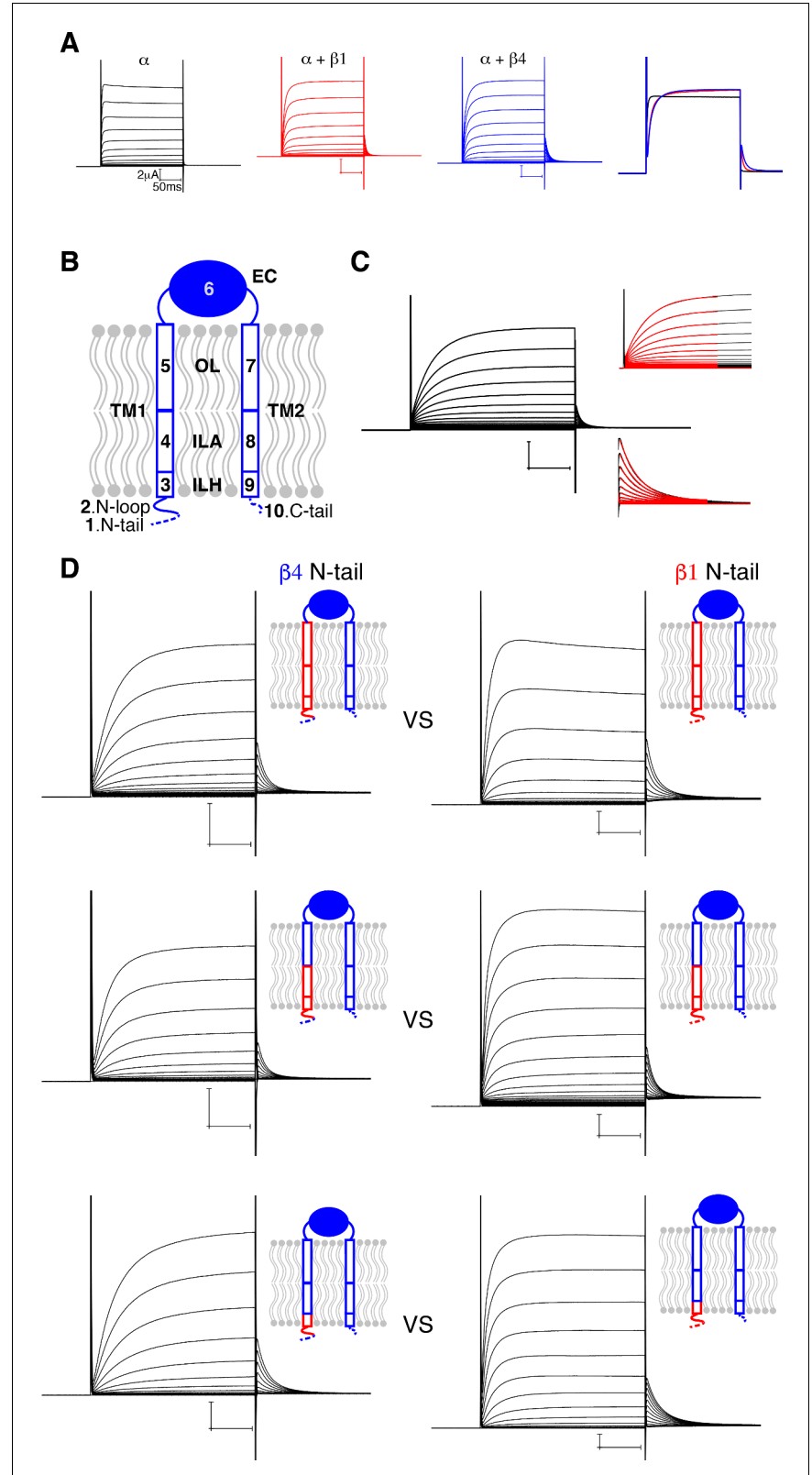

**Figure 4.** Influence of β4 N-terminus on Slo1 gating. (**A**) Voltage-dependent channel activation of the human Slo1 channel alone and co-expressed with the β1 or β4 subunit. Representative current traces recorded using two-electrode voltage clamp (TEVC) are shown. Recording buffer: HEPES 5 mM, KCl 98 mM, CaCl₂ 0.3 mM, and MgCl₂ 1 mM. Voltage protocol: holding potential 0 mV, step from −80 to 120 mV in 10 mV incremental steps, step
*Figure 4 continued on next page*

*Figure 4 continued*

back to 40 mV. A superposition of one single sweep (stepping to 100 mV) from the left three recordings is shown on the right. (B) A schematic drawing of the β4 subunit showing the 10 regions divided based on the atomic structure. (C) Quantification of channel activation and deactivation kinetics by fitting with a single exponential function (see Materials and methods). (D) Comparison of three pairs of mutants demonstrated that the nature of N-tail is correlated with the activation kinetics: the presence of β4 N-tail (left column) results in slower activation kinetics than β1 N-tail (right column). Schematic drawing of each corresponding mutant is shown next to the representative current traces. Voltage protocols are the same as in panel (A).

The online version of this article includes the following figure supplement(s) for figure 4:

**Figure supplement 1.** Identification of the β4 regions with critical functional effects with TEVC.

Activation and deactivation kinetics were quantified by fitting a single exponential function to the current time course (*Figure 4C*).

*Figure 4—figure supplement 1A* shows the plot of activation time constants ($\tau$_on) of wild-type (wt) and all mutants in an ascending order from left to right. Based on their difference from the β4 or β1 wt values, these mutants are subdivided into four categories ranging from $\tau$_on less than one half that of β4 wt (i.e. faster rates) to greater than two times β4 wt (i.e. slower rates). Deactivation

**Table 2.** List of the β4 mutants for TEVC studies, related to *Figure 4*.

| ShortName | β4 sequence | β1 sequence | ShortName | β4 sequence | β1 sequence |
|---|---|---|---|---|---|
| Slo1 | N/A | N/A | m27 | 1-10,20-190,206-210 | 10-18,178-191 |
| β1 | N/A | 1-191 | m28 | 11-193 | 1-9,181-191 |
| β4 | 1-210 | N/A | m29 | 11-198 | 1-9,181-186 |
| m2 | 7-210 | N/A | m30 | 49-210 | 1-47 |
| m3 | 1-205 | N/A | m31 | 1-163 | 151-191 |
| m4 | 10-210 | N/A | m32 | 1-10,49-210 | 10-47 |
| m5 | 14-210 | 1-12 | m33 | 1-163,206-210 | 151-191 |
| m6 | 1-34,40-210 | 34-38 | m34 | 1-10,49-163 | 10-47,151-191 |
| m7 | 1-190,197-210 | 178-183 | m35 | 1-28,49-210 | 28-47 |
| m8 | 1-10,14-210 | 10-12 | m36 | 1-163,181-210 | 151-167 |
| m9 | E9A | N/A | m37 | 1-19,30-210 | 19-28 |
| m10 | 1-18,20-210 (R19C) | 18 | m38 | 1-180,191-210 | 168-177 |
| m11 | R19L | N/A | m39 | 1-10,30-210 | 10-28 |
| m12 | E14A | N/A | m40 | 1-180,206-210 | 168-191 |
| m13 | 1-42,44-210 (A43Y) | 42 | m41 | 1-10,20-210 | 10-18 |
| m14 | 1-164,168-210 | 152-154 | m42 | 1-190,206-210 | 178-191 |
| m16 | 7-205 | N/A | m43 | 11-210 | 1-9 |
| m17 | 1-42,44-164,168-210 | 42,152-154 | m44 | replace β4EC (49-166) with "GGSGGGSG" | N/A |
| m18 | 1-11,13-14,16-18,20-210 (E12R, D15T, R19C) | 11,14,18 | m45 | replace β1EC (41-153) with "GGSGGGSG" | N/A |
| m19 | 1-48,164-210 | 48-150 | m46 | replace β1EC (48-153) with "GGSGGGSG" | N/A |
| m20 | 1-45,167-210 | 45-153 | m47 | 1-47,151-210 | 49-163 |
| m21 | 49-163 | 1-47,151-191 | m48 | 1-44,154-210 | 46-166 |
| m22 | 1-10,49-163,206-210 | 10-47,151-191 | m49 | 1-6,49-210 | 6-47 |
| m23 | 1-10,49-163,196-210 | 10-47,151-182 | m50 | 1-6,30-210 | 6-28 |
| m24 | 1-28,49-163,181-210 | 28-47,151-167 | m51 | 1-2,30-210 | 2-28 |
| m25 | 1-19,30-180,191-210 | 19-28,168-177 | m52 | 1-6,20-210 | 6-18 |
| m26 | 1-10,30-180,206-210 | 10-28,168-191 | m53 | 1-2,20-210 | 2-18 |

time constants ($\tau$_off) were plotted in a similar fashion (*Figure 4—figure supplement 1B*). Mutants with faster activation rates were distributed throughout the β4 structure, while those with significantly slower activation rates were distributed with some bias towards the inner leaflet region of TM1, the N-loop, and the N-tail (*Figure 4—figure supplement 1A and C*, *Table 2*). While mutations affecting $\tau$_on were not strongly correlated with effects on $\tau$_off, those causing the greatest slowing of deactivation also involved the N-terminus of TM1, the N-loop and N-tail (*Figure 4—figure supplement 1B* and *Table 2*).

Although not observed in the cryo-EM structure, the N-tail plays a definitive role in determining activation kinetics. Specifically, mutants containing the N-tail of β4 activate more slowly than those with the N-tail of β1. This effect is demonstrated in *Figure 4D*, which shows the influence of the β4 versus the β1 N-tail independent of mutations elsewhere in the body of the β subunit. The importance of the β subunit N-terminus to gating has already been proposed (*Castillo et al., 2015*). We also observe that removal of the N-tail results in a modest increase in the rate of activation (*Figure 4—figure supplement 1* and *Table 2*). In the open structure of the Slo1 α-β4 complex, the N-loop of β4 is located next to the S6-RCK1 linker of α. It would seem likely that the close proximity of this N-terminal region, including the N-tail of the β4 subunit, to an important gating region of the α subunit, the S6-RCK1 linker, somehow underlies the effects on activation that we observe.

## Ca²⁺-induced Pore Opening in the Context of the β4 Subunit

$Ca^{2+}$ binding to two unique $Ca^{2+}$ binding sites per α subunit produces protein conformational changes that splay the S6 (inner) helices, causing a wide pathway for ion diffusion to open between the cytoplasm and the transmembrane pore (*Figure 5A*). The $Ca^{2+}$ binding sites are located within the cytoplasmic 'gating ring', which is centered on the four-fold channel axis just beneath the transmembrane pore (*Figures 1B* and *5B*). When $Ca^{2+}$ binds, a domain of the gating ring closest to the membrane, called the RCK1 N-lobe, expands away from the channel's central axis; expansion of all four RCK1 N-lobes, one from each subunit, produces a radial expansion, as shown (*Figure 5B*). Since the RCK1 N-lobes are connected directly to the S6 helices via the S6-RCK1 linkers, their expansion produces pore opening. Structural details of the $Ca^{2+}$ binding site reorganization upon $Ca^{2+}$ binding, RCK1 N-lobe expansion and pore opening observed in human Slo1 are essentially the same as those observed previously in acSlo1 (*Hite et al., 2017*; *Tao et al., 2017*). Thus, the intricacies of $Ca^{2+}$-mediated pore opening are conserved among Slo1 channels from invertebrates to mammals.

The $Ca^{2+}$-bound structures of the human Slo1 α subunit determined in the absence and presence of the β4 subunit are essentially identical to each other (RMSD 2.0 Å) (*Figure 5—figure supplement 1A*). Likewise, the two $Ca^{2+}$-free structures of the α subunit (± β4 subunit) are also the same within the accuracy of our measurements (RMSD 1.1 Å) (*Figure 5—figure supplement 1B*). Contrasting the large $Ca^{2+}$-mediated conformational changes that occur within the α subunit pore (*Figure 5A*) and gating ring (*Figure 5B*) with the absence of conformational change in the β4 subunit (*Figure 5C*) gives the impression that the β4 subunits encage the α subunit channel without interfering with its $Ca^{2+}$-mediated gating conformational changes. Within the outer leaflet of the membrane even lipid molecules that mediate interactions between the α and β subunits remain unperturbed (*Figure 5D and E*). Within the inner leaflet of the membrane, where large gating conformational changes in the α subunit occur (*Figure 5A and F–H*), interfaces between α and β subunits – especially surrounding the more centrally located TM1 of the β subunit – undergo change. This N-terminal segment of TM1 corresponds to the region where mutations had the largest influence on gating kinetics (*Figure 5F–H* and *Figure 4—figure supplement 1*).

In functional studies, β subunits alter the apparent $Ca^{2+}$ sensitivity of gating. But the sensitivity changes are not thought to reflect changes in intrinsic $Ca^{2+}$ affinity for the binding sites, but instead are proposed to reflect indirect effects that β subunits have on α subunit conformational changes that are coupled to $Ca^{2+}$ binding (*Cox and Aldrich, 2000*; *Nimigean and Magleby, 2000*; *Bao and Cox, 2005*; *Savalli et al., 2007*; *Contreras et al., 2012*). The structures support this proposal because the $Ca^{2+}$ binding sites are unaltered by the presence of β subunits (*Figure 5—figure supplement 1C–F*). This study only determined structures at very low and very high $Ca^{2+}$ concentrations. Had we determined structures at intermediate $Ca^{2+}$ concentrations we suspect that we would observe the same closed and open conformations with a distribution accounting for their relative probabilities weighted according to the $Ca^{2+}$ concentration (*Hite and MacKinnon, 2017*). We expect that the 'weighting function' would be altered by the presence of β subunits. The structural

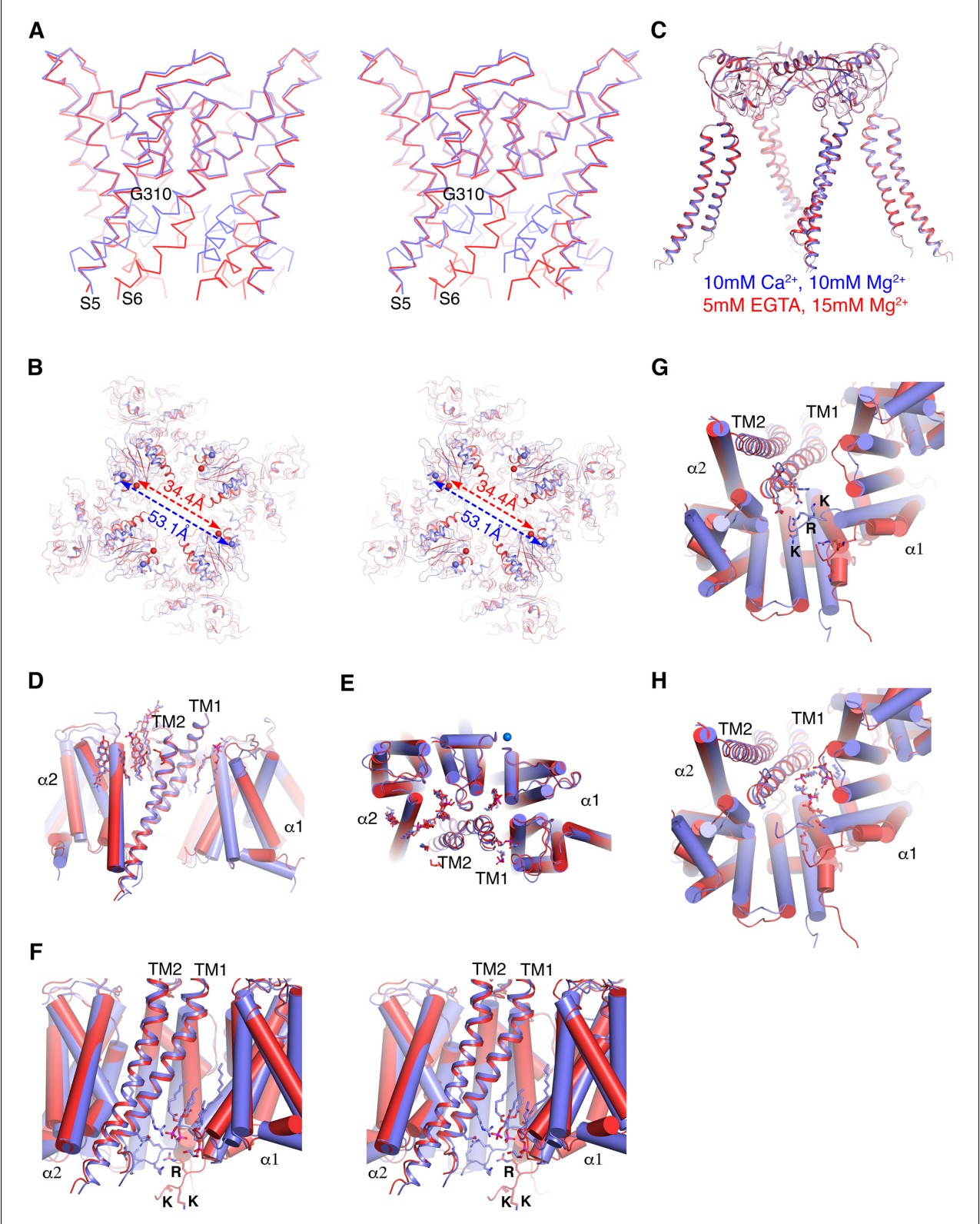

**Figure 5.** Ca$^{2+}$gating mechanism of human Slo1 in the absence and presence of β4 subunit. (**A**) Ca$^{2+}$-induced conformational changes in the pore domain of human Slo1-β4 channel complex. Superposition of the pore domain in the absence (red Cα trace) and presence (blue Cα trace) of Ca$^{2+}$ is shown in stereo. The gating hinge residue G310 on inner helix is labeled. (**B**) Ca$^{2+}$-induced conformational changes in the gating ring of human Slo1-β4 channel complex. Superposition of the gating ring (aligning the RCK2 domain) in the absence (red) and presence (blue) of Ca$^{2+}$ is shown in stereo. The

*Figure 5 continued on next page*

*Figure 5 continued*

spheres indicate the position of Cα atoms of Gly334 at the beginning of RCK1 domain. Distances between the Cα atoms of Gly334 on opposing RCK1 subunits are labeled. (C) $Ca^{2+}$ produced essentially no conformational changes in the β4 tetramer. Superposition of the β4 tetramer in the absence (red) and presence (blue) of $Ca^{2+}$ is shown. (D, E) $Ca^{2+}$ produced minimal conformational changes in the Slo1-β4 interfaces near the membrane outer leaflet, including the positions of lipid molecules, viewed parallel to membrane (D) or from the extracellular side (E). Superposition of the Slo1-β4 channel complex in the absence (red) and presence (blue) of $Ca^{2+}$ is shown, aligning the transmembrane domain. For clarity, only one β4 subunit (as ribbons) and the two interacting Slo1 subunits (α1 and α2) (as cylinders) are shown. Lipids at the Slo1 and β4 TM1 outer leaflet interface are shown as sticks. $K^+$ ions in the selectivity filter are show as marine spheres (E). (F) $Ca^{2+}$-induced conformational changes in the Slo1-β4 interfaces near the membrane inner leaflet. Superposition of the Slo1-β4 channel complex in the absence (red) and presence (blue) of $Ca^{2+}$ is shown in stereo, viewed parallel to the membrane, aligning the transmembrane domain. Sidechains of the β4 TM1 facing the S6-RCK1 linker as well as the three positively charged residues on S6-RCK1 linker ('RKK') are shown as sticks. Lipids at the Slo1 and β4 TM1 inner leaflet interface in the $Ca^{2+}$-bound state are also shown as sticks. (G) Superposition of the Slo1-β4 channel complex in the absence (red) and presence (blue) of $Ca^{2+}$ viewed from the intracellular side, aligning the transmembrane domain. Color and representation schemes are the same as in panel (F). Sidechains of the β4 TM1 facing the S6-RCK1 linker as well as the three positively charged residues on S6-RCK1 linker ('RKK') are shown as sticks. (H) The same superposition as in panel (G) with lipids at the Slo1 and β4 TM1 inner leaflet interface in the $Ca^{2+}$-bound state (blue) and $Ca^{2+}$-free state (red) shown as sticks.

The online version of this article includes the following figure supplement(s) for figure 5:

**Figure supplement 1.** β4 has minimal impact on the $Ca^{2+}$-induced open and closed conformations of hsSlo1.

---

data support the idea that β subunits stabilize or destabilize conformational states of the α subunits rather than creating new states. The RCK1 N-lobe and S6-RCK1 linker region of the α subunit is one likely region where β subunits exert their effects. We also must point out that the structures determined in this study, corresponding to a cycle relating β subunits to $Ca^{2+}$ binding (*Figure 1A*), are silent on direct effects that the β subunits could have on different conformations of the voltage sensors. It is very possible that yet unknown structural changes brought about by a transmembrane electric field could alter interactions between β subunits and the voltage sensors of the α subunits. Direct stabilization of different conformations of voltage sensors by β subunits remains a distinct mechanistic possibility (*Bao and Cox, 2005*; *Savalli et al., 2007*; *Contreras et al., 2012*).

## Structural basis for modification of Slo1 toxin sensitivity by β subunits

Certain protein toxins from scorpion venoms inhibit $K^+$ channels by plugging their extracellular pore entryway (*Anderson et al., 1988*; *MacKinnon and Miller, 1988*). Our understanding of these toxins' mechanism of action as pore blockers comes mainly from studies of charybdotoxin (CTX) and Iberiotoxin (IbTX) inhibition of Slo1 channels (*Candia et al., 1992*; *Giangiacomo et al., 1992*; *Miller, 1995*). The large conductance of Slo1 has enabled the measurement of toxin association and dissociation events in single channel recordings (*MacKinnon and Miller, 1988*). About 30 years ago a Slo1 with altered toxin binding properties was discovered (*Reinhart et al., 1989*). The altered properties were later attributed to the presence of β4 subunits (*Behrens et al., 2000*; *Brenner et al., 2000*; *Meera et al., 2000*; *Weiger et al., 2000*; *Lippiat et al., 2003*). We now know that different β subunits confer unique pharmacological profiles to the Slo1 channel (*Behrens et al., 2000*; *Brenner et al., 2000*; *Lippiat et al., 2003*). Studies with mutant β subunits showed that toxins are most sensitive to the composition of the EC domain (*Meera et al., 2000*).

The α-β4 structure provides a simple explanation for the general observation that β subunits slow the rates of association and dissociation of toxins with the channel: the EC domain crown limits access to the toxin binding site. Using the crystal structure of CTX bound to the Kv1.2-Kv2.1 paddle chimera channel to model CTX binding to Slo1, we conclude that there is adequate space for toxin binding within the cage beneath the crown, but that CTX has restricted access across the central opening to reach its binding site (*Figure 6A–C* and *Figure 6—figure supplement 1*) (*Banerjee et al., 2013*). The toxin is elongated along one axis and likely has to orient with its long axis parallel to the 4-fold axis to pass through the opening on the top of the crown. Optimal orientation will still require amino acid sidechain adjustments on both the toxin and β subunits to permit passage. Restricted access rationalizes a reduced association rate (compared to a Slo1 channel without β subunits) if the probability of successful encounter is reduced (*Schurr, 1970*). Analogously, a reduced dissociation rate would reflect a reduced probability of toxin exit. If the toxin releases from its binding site and tumbles within the cage beneath the crown, rebinding many times before finally exiting, the resultant effect would be purely kinetic (the assumption here being that release and

rebinding without dissociation from the cage would be too rapid to resolve as dissociation events in channel recordings). If, on the other hand, toxin is stabilized on its binding site by the β subunit, then equilibrium effects (i.e. effects on affinity) would also be expected. Data suggest that both kinetic (i.e. rate changes that could be modeled through a barrier height) and equilibrium effects (i.e. affinity changes that could be modeled through the depth of a toxin binding well) occur (*Behrens et al., 2000*; *Brenner et al., 2000*; *Meera et al., 2000*; *Lippiat et al., 2003*).

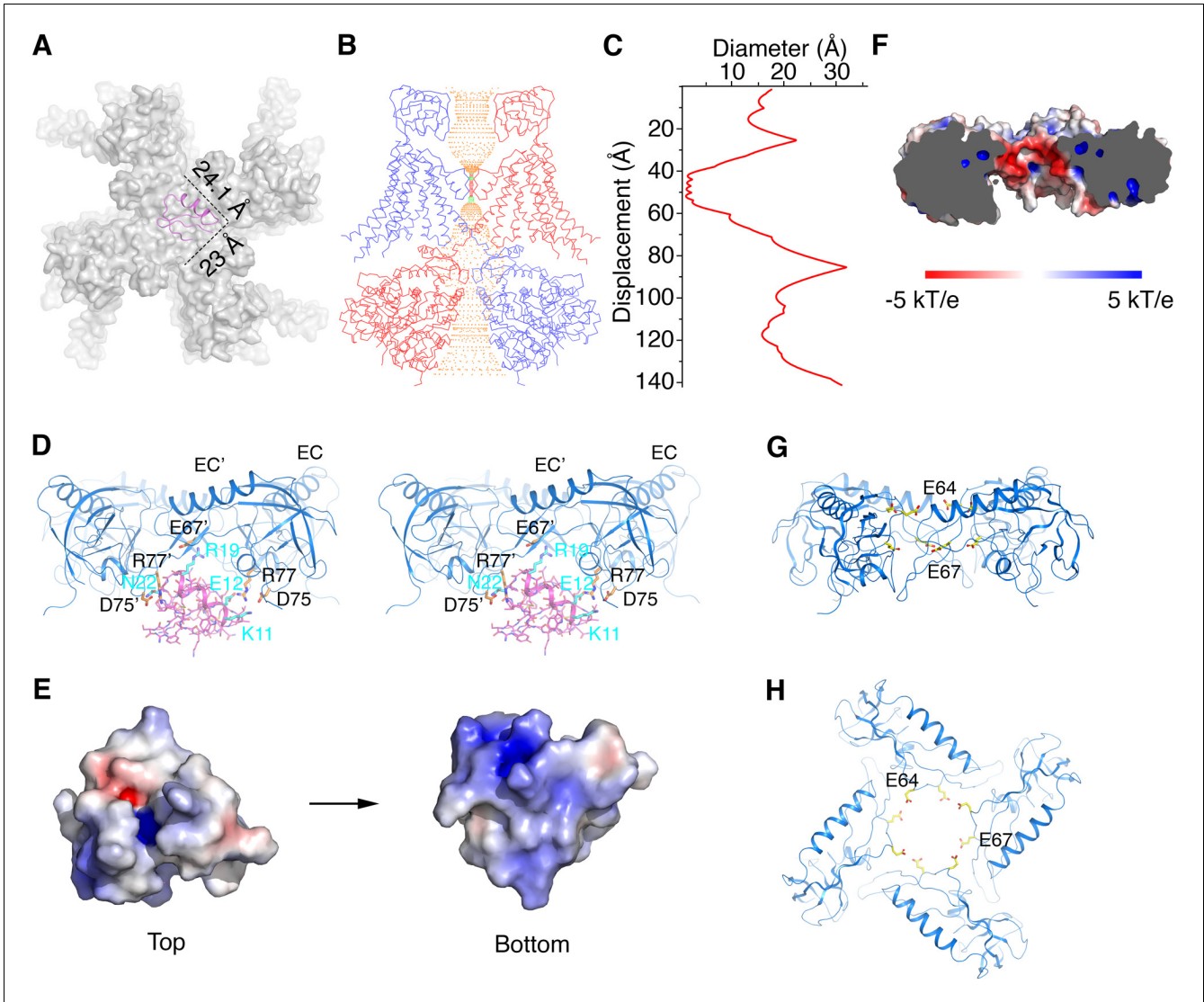

**Figure 6.** Structural basis for modification of Slo1 toxin sensitivity by β subunits. (**A**) CTX docked onto the Slo1 channel based on the crystal structure of the CTX-Kv1.2–2.1 paddle chimera (PDB 4JTA). Only the transmembrane domain of Slo1 is shown (gray surface) and CTX is shown as pink ribbons. Dimensions of CTX in the plane parallel to membrane are indicated. (**B**) Central conduction pore of the open Slo1-β4 channel complex generated with Hole (*Smart et al., 1996*). For clarity, only two opposing subunits of Slo1 and β4 are shown (blue and red Cα traces). Pore radius: red,<1.15 Å; green, 1.15 to 2.30 Å; orange,>2.30 Å. (**C**) Diameter of the central pore. The van der Waals radius is plotted against the distance along the pore axis. (**D**) The EC domain tetramer of β4 provides potential new toxin binding sites. The EC tetramer is shown as blue ribbons. CTX is shown as pink sticks and ribbons. Potential new CTX binding sites on the EC domain and the corresponding interacting residues on CTX are shown as sticks and colored according to atom type. (**E**) Electrostatic surface potential of CTX viewed from the extracellular side ('Top') or the opposite side ('Bottom'), calculated with APBS. (**F**) Negatively charged inner surface of the central pore formed by the EC tetramer of β4 subunits, calculated with APBS. (**G, H**) Two rings of negatively charged residues E64 and E67 on the EC domain of β4, facing the central pore axis, viewed parallel to the membrane (**G**) or from the extracellular side (**H**). EC domains are shown as blue ribbons. Sidechains of E64 and E67 are shown as sticks and colored according to atom type. The online version of this article includes the following figure supplement(s) for figure 6:

**Figure supplement 1.** CTX docked onto the Ca²⁺-bound hsSlo1-β4 channel complex.

A number of prior experiments and conclusions are consistent with the α-β4 structure. In 1994 Garcia and colleagues demonstrated that K69 on β1 (corresponding to R77 on β4) can be cross-linked to CTX (*Knaus et al., 1994a*). R77 is located inside the cage formed by the EC domain crown, facing the backside of CTX (i.e. the side opposite the pore's selectivity filter) (*Figure 6D*). Scorpion toxins generally are electropositive on their surface: CTX and IbTX contain a total of 8 and 7 positively charged residues, respectively (*Figure 6E* and *Figure 2—figure supplement 1C*). Garcia et. al. speculated that certain negatively charged residues within the large extracellular loop of the β subunit attract CTX to its binding site on the α subunit (*Hanner et al., 1998*). In line with this hypothesis, the inner surface of the cage formed by the EC domain crown is electronegative due to the presence of two rings of negatively charged residues – E64 and E67 (*Figure 6F–H*). These positions are almost strictly conserved as negatively charged residues (E or D) in β1 to β4 (*Figure 2—figure supplement 1A*), which would create electrostatic interactions between the toxin and the β subunit. Mutations made on the back side of CTX have little effect on affinity for Slo1 without β subunits (*Park and Miller, 1992*). The structural model predicts that some of the same CTX mutations are likely to have an effect in the presence of β subunits (*Figure 6D*).

## Discussion

The ion channel composed of the α subunit of Slo1 forms a functional $Ca^{2+}$- and voltage-gated $K^+$ channel. In different cells within an organism the functional properties of Slo1 are not all the same: variations in the kinetics of channel opening and closing and in the apparent $Ca^{2+}$ sensitivity are observed. These variations exist because the functional properties of Slo1 channels are 'tuned' through RNA splicing, posttranslational modifications, and assembly of the Slo1 α subunit with auxiliary β or γ subunits. In this study we for the first time visualize the assembly of the Slo1 α subunit with an auxiliary subunit, the β4 subunit. The structure offers a mechanistic picture explaining certain functional properties, such as the influence of β subunits on toxin interactions with Slo1. The structure also implies that the N-terminus of the β subunit TM1 and the segment of β4 preceding TM1 influence channel gating through interactions with the gating ring RCK1 N-lobe and the S6-RCK1 linker. As is often the case, because functional properties of a protein can be sensitive to atomic displacements smaller than the resolution of a structure, many functional effects of β subunits are not easily explained by the structure. For the unexplained properties, the structure still serves as an essential starting point for future understanding.

The Slo1 β subunits are one example of relatively small (i.e. TM segments not much larger than a few lipid molecules) membrane proteins that regulate the function of ion channels. Slo1 also has γ subunits, the KCNQ (Kv7) voltage-dependent $K^+$ channels co-assemble with single membrane-spanning KCNE subunits, Kv4 co-assembles with dipeptidyl aminopeptidase-like (DPPL) proteins and voltage-dependent $Na^+$ (Nav) channels are also associated with auxiliary subunits (*Pongs and Schwarz, 2010*; *Calhoun and Isom, 2014*; *Zhang and Yan, 2014*). These small transmembrane proteins are generally multifunctional, affecting various aspects of the larger target protein including trafficking/surface expression, biophysical properties, pharmacological profiles, and assembly into functional complexes (*Pongs and Schwarz, 2010*; *Zickermann et al., 2010*; *Calhoun and Isom, 2014*). DPPL for Kv4 has been implicated in binding to components of the extracellular matrix through its extracellular domain (*Pongs and Schwarz, 2010*), and the Nav β subunit extracellular domain plays a crucial role in cell adhesion and migration (*Calhoun and Isom, 2014*). Whether the structured crown of Slo1 β subunits (sequence similarity virtually guarantees a similar extracellular structure in β1-β3 as well) interacts with extracellular matrix proteins to localize Slo1 channels to specific regions of a cell, perhaps regions of contact with other cells, is still unknown, but seems like a good possibility.

Another idea we have when looking at the α-β4 Slo1 structure is inspired by the location of the β subunit TM helices. When they encircle the α subunit, they must displace lipid molecules. As shown in the structures, certain lipid molecules are bound at the α/β subunit interfaces, but the presence of β subunits necessitates the displacement of some lipids. We know that the function of many ion channels is very sensitive to the membrane lipid composition (*Heginbotham et al., 1998*; *Valiyaveetil et al., 2002*; *Schmidt et al., 2006*; *Schmidt et al., 2009*; *Cheng et al., 2011*). Thus, we consider it perhaps useful to view these small transmembrane spanning proteins as membrane

components that, like lipids, by altering the chemical and physical properties of the surrounding membrane, alter the function of the ion channel.

# Materials and methods

## Key resources table

| Reagent type (species) or resource | Designation | Source or reference | Identifiers | Additional information |
|---|---|---|---|---|
| Gene (*Homo sapiens*) | hsSlo1 (human_KCNMA1) | synthetic | accession: Q12791.2 GI: 46396283 | synthesized at GeneWiz |
| gene (*Homo sapiens*) | hsbeta4 (human_KCNMB4) | synthetic | accession: NP_055320.4 GI: 26051275 | synthesized at GeneWiz |
| gene (*Homo sapiens*) | hsbeta1 (human_KCNMB1) | synthetic | accession: Q16558.5 GI: 292495100 | synthesized at GeneWiz |
| Recombinant DNA reagent | pEG BacMam | DOI: 10.1038/nprot.2014.173 | | |
| Recombinant DNA reagent | pGEM | https://www.add gene.org/vector-database/2835/ | | |
| Cell line (*Homo sapiens*) | HEK293S GnTI⁻ | ATCC | CRL-3022 | cells purchased from ATCC and we have now confirmed there is no mycoplasma contamination |
| Cell line (*Homo sapiens*) | HEK293T | ATCC | CRL-3216 | cells purchased from ATCC and we have now confirmed there is no mycoplasma contamination |
| Cell line (*Spodoptera frugiperda*) | Sf9 | ATCC | CRL-1711 | cells purchased from ATCC and we have now confirmed there is no mycoplasma contamination |
| Strain, strain background (*Escherichia coli*) | DH10Bac | ThermoFisher | 10361012 | MAX Efficiency DH10Bac Competent Cells |
| Biological sample (*Xenopus laevi*) | oocyte | *Xenopus laevi* | | |
| Chemical compound, drug | Freestyle 293 medium | Gibco | 12338018 | |
| Chemical compound, drug | sf-900 II SFM medium | Gibco | 10902088 | |
| Chemical compound, drug | 2,2-didecylpropane -1,3-bis-β-D-malto pyranoside (LMNG) | Anatrace | NG310 | |
| Chemical compound, drug | Cholesteryl hemisuccinate (CHS) | Anatrace | CH210 | |
| Chemical compound, drug | Digitonin | Sigma-Aldrich | D141 | |
| Chemical compound, drug | Cellfectin II | Invitrogen | 10362100 | |
| Chemical compound, drug | FuGENE HD transfection reagent | Promega | E2312 | |
| Chemical compound, drug | Collegenase type II | Gibco | 17107–0125 | |

*Continued on next page*

*Continued*

| Reagent type (species) or resource | Designation | Source or reference | Identifiers | Additional information |
|---|---|---|---|---|
| Chemical compound, drug | Gentamicin sulphate | Sigma-Aldrich | A0752 | |
| Commercial assay or kit | CNBr-activated sepharose beads | GE Healthcare | 17043001 | |
| Commercial assay or kit | Superose 6, 10/300 GL | GE Healthcare | 17517201 | |
| Commercial assay or kit | mMESSAGE mMACHINE T7 Transcription Kit | ThermoFisher | AM1344 | |
| Commercial assay or kit | AmpliCap-Max T7 high yield message maker kit | CELLSCRIPT | C-ACM04037 | |
| Commercial assay or kit | R1.2/1.3 400 mesh Au holey carbon grids | Quantifoil | 1210627 | |
| Software, algorithm | SerialEM | DOI: 10.1016/j.jsb.2005.07.007 | http://bio3d.colorado.edu/SerialEM | |
| Software, algorithm | MotionCor2 | DOI: 10.1038/nmeth.4193 | https://msg.ucsf.edu/software | |
| Software, algorithm | Gctf | DOI: 10.1016/j.jsb.2015.11.003 | https://www.mrc-lmb.cam.ac.uk/kzhang/ | |
| Software, algorithm | Gautomatch | other | https://www.mrc-lmb.cam.ac.uk/kzhang/ | |
| Software, algorithm | cryoSPARC | DOI: 10.1038/nmeth.4169 | http://www.cryosparc.com | |
| Software, algorithm | RELION-3 | DOI: 10.7554/eLife.18722 | http://www2.mrc-lmb.cam.ac.uk/relion | |
| Software, algorithm | FrealignX | DOI: 10.1016/j.jsb.2013.07.005 | http://grigoriefflab.janelia.org/frealign | |
| Software, algorithm | COOT | DOI: 10.1107/S0907444910007493 | https://www2.mrc-lmb.cam.ac.uk/personal/pemsley/coot/ | |
| Software, algorithm | PHENIX | DOI: 10.1107/S2059798318006551 | https://www.phenix-online.org | |
| Software, algorithm | UCSF Chimera | DOI: 10.1002/jcc.20084 | https://www.cgl.ucsf.edu/chimera | |
| Software, algorithm | Pymol | PyMOL Molecular Graphics System, Schrödinger, LLC | http://www.pymol.org | |
| Software, algorithm | HOLE | DOI: 10.1016/s0263-7855(97)00009-x | http://www.holeprogram.org | |
| Software, algorithm | pClamp | Axon Instruments, Inc | | |

## Cloning, Expression and Purification

HsloM3 (GI: 507922, here referred to as human Slo1 or hsSlo1) was generously provided by Ligia Toro in a pcDNA3 vector and served as the template for subcloning. To improve the biochemical stability of hsSlo1, we excluded the very C-terminal 57 residues at the level of molecular biology. Specifically, a gene fragment encoding residues 1–1056 of hsSlo1 was subcloned into a modified pEG BacMam vector (*Goehring et al., 2014*). The resulting protein has green fluorescent protein (GFP) and a 1D4 antibody recognition sequence (TETSQVAPA) on the C-terminus, separated by a PreScission protease cleavage site (SNSLEVLFQ/GP). This truncated construct, denoted hsSlo1$_{EM}$, was used in all the experiments of this manuscript.

Synthetic gene fragments (Genewiz) encoding full length human β4 (GI: 26051275, residues 1–210) and β1 (GI: 4758626, residues 1–191) subunit of Slo1 were subcloned into a similarly modified

pEG BacMam vector with mCherry and a deca-histidine affinity tag (mCherry-His10) replacing the GFP-1D4 fragment.

HsSlo1 was either expressed alone or co-expressed with the β4 subunit in HEK293S GnTI⁻ cells using the BacMam method (*Goehring et al., 2014*). Briefly, bacmid carrying hsSlo1$_{EM}$ or β4 subunit was generated by transforming *E. coli* DH10Bac cells with the corresponding pEG BacMam construct according to the manufacturer's instructions (Bac-to-Bac; Invitrogen). Baculoviruses were produced by transfecting *Spodoptera frugiperda* Sf9 cells with the bacmid using Cellfectin II (Invitrogen). Baculoviruses, after two rounds of amplification, were used for cell transduction. Suspension cultures of HEK293S GnTI⁻ cells were grown at 37°C to a density of ~3×10⁶ cells/ml. For expression of hsSlo1 alone, cell culture was infected with 15% (v:v) of hsSlo1$_{EM}$ baculovirus. For co-expression of hsSlo1 and β4 subunit, cell culture was infected with 5% (v:v) hsSlo1$_{EM}$ plus 15% (v:v) of β4 baculoviruses to initiate the transduction. After 20 hr, 10 mM sodium butyrate was supplemented and the temperature was shifted to 30°C. Cells were harvested ~40 hr after the temperature switch.

For the Ca$^{2+}$-bound hsSlo1 protein sample, cells were gently disrupted by stirring in a hypotonic solution containing 10 mM Tris-HCl pH 8.0, 3 mM dithiothreitol (DTT), 1 mM EDTA supplemented with protease inhibitors including 0.1 µg/ml pepstatin A, 1 µg/ml leupeptin, 1 µg/ml aprotinin, 0.1 mg/ml soy trypsin inhibitor, 1 mM benzamidine, 0.1 mg/ml 4-(2-Aminoethyl) benzenesulfonyl fluoride hydrochloride (AEBSF) and 1 mM phenylmethysulfonyl fluoride (PMSF). Cell lysate was then centrifuged for 30 min at 30,000 g and pellet was homogenized in a buffer containing 20 mM Tris-HCl pH 8.0, 320 mM KCl, 10 mM CaCl$_2$, 10 mM MgCl$_2$ supplemented with protease inhibitors including 0.1 µg/ml pepstatin A, 1 µg/ml leupeptin, 1 µg/ml aprotinin, 0.1 mg/ml soy trypsin inhibitor, 1 mM benzamidine, 0.1 mg/ml AEBSF and 0.2 mM PMSF. The lysate was extracted with 10 mM lauryl maltose neopentyl glycol (LMNG) and 2 mM cholesteryl hemisuccinate (CHS) for an hour with stirring and then centrifuged for 40 min at 30,000 g. Supernatant was added to GFP nanobody-conjugated affinity resin (CNBr-activated Sepharose 4B resin from GE Healthcare) pre-equilibrated with wash buffer (20 mM Tris-HCl pH 8.0, 450 mM KCl, 10 mM CaCl$_2$, 10 mM MgCl$_2$, 0.005% digitonin (Sigma), 0.1 mg/ml 1-palmitoyl-2-oleoyl-sn-glycero-3-phosphoethanolamine (POPE): 1-palmitoyl-2-oleoyl-glycero-3-phosphocholine (POPC): 1-palmitoyl-2-oleoyl-sn-glycero-3-phosphate (POPA) 5:5:1 (w:w:w), 0.1 µg/ml pepstatin A, 1 µg/ml aprotinin and 0.1 mg/ml soy trypsin inhibitor) (*Fridy et al., 2014*). The suspension was mixed by nutating for ~2 hr. Beads were first washed with 10 column volumes of wash buffer in batch mode and then collected on a column by gravity, washed with another 20 column volumes of wash buffer. The protein was then digested on resin with PreScission protease (~20:1 w:w ratio) overnight with gentle rocking. Flow-through was then collected, concentrated and further purified on a Superose-6 size exclusion column in 20 mM Tris-HCl pH 8.0, 450 mM KCl, 10 mM CaCl$_2$, 10 mM MgCl$_2$, 0.1 µg/ml pepstatin A, 1 µg/ml aprotinin, 0.005% digitonin and 0.05 mg/ml POPE:POPC:POPA 5:5:1 (w:w:w). All purification procedures were carried out either on ice or at 4°C. The peak fractions corresponding to the tetrameric Slo1 channel was concentrated to about 7 mg/ml and used for preparation of cryo-EM sample grids.

The Ca$^{2+}$-free hsSlo1 protein sample was prepared in a similar fashion with 5 mM EGTA and 15 mM MgCl$_2$ substituting 10 mM CaCl$_2$ and 10 mM MgCl$_2$. For the Ca$^{2+}$-bound and Ca$^{2+}$-free hsSlo1-β4 complex protein samples, 1% Digitonin was used for extraction and final protein concentration was at ~8.5 mg/ml.

For confirmative studies of the two predicted N-glycosylation sites on human β4, single and double N2Q mutants were made and confirmed by sequencing (Genewiz). The hsSlo1$_{EM}$-β4 glycosylation mutants complex were expressed and purified the same as hsSlo1$_{EM}$-β4 wild-type. The purified proteins were analyzed with tandem mass spectrometry (ms/ms) at the Proteomics Resource Center of the Rockefeller University.

## Cryo-EM grid preparation and imaging

3.5 µl of purified protein sample was pipetted onto glow-discharged Quantifoil Au 400 mesh, R 1.2/1.3 holey carbon grids (Quantifoil). Grids were blotted for 4 s with a blotting force of 1 and humidity of 100% and flash frozen in liquid-nitrogen-cooled liquid ethane using a FEI Vitrobot Mark IV (FEI). Grids were then transferred to a FEI Titan Krios electron microscope operating at an acceleration voltage of 300 keV. Images were recorded in an automated fashion on a Gatan K2 Summit detector (Gatan) set to super-resolution mode using SerialEM (*Mastronarde, 2005*). Images of Ca$^{2+}$-bound (open) hsSlo1-β4 complex were recorded with an energy filter of 20 eV at a super-

resolution pixel size of 0.52 Å and defocus range of 0.7 to 2.0 µm, for 10 s with a subframe exposure time of 200 ms in a dose of approximately eight electrons per pixel per second (a total accumulated dose of approximately 74 electrons per $Å^2$ over 50 subframes or approximately 1.5 electrons per $Å^2$ per subframe). Images of $Ca^{2+}$-free (closed) hsSlo1-β4 complex were recorded at a super-resolution pixel size of 0.65 Å and defocus range of 0.7 to 2.0 µm, for 15 s with a subframe exposure time of 300 ms in a dose of approximately 10 electrons per pixel per second (a total accumulated dose of approximately 89 electrons per $Å^2$ over 50 subframes or approximately 1.78 electrons per $Å^2$ per subframe). Images of $Ca^{2+}$-bound (open) hsSlo1 and $Ca^{2+}$-free (closed) hsSlo1 were recorded at a super-resolution pixel size of 0.65 Å and defocus range of 0.8 to 2.4 µm for 15 s with a subframe exposure time of 300 ms in a dose of approximately 10 electrons per pixel per second (a total accumulated dose of approximately 89 electrons per $Å^2$ over 50 subframes or approximately 1.78 electrons per $Å^2$ per subframe).

## Image processing and map calculation

Dose-fractionated super-resolution images were 2×2 down sampled by Fourier cropping for motion correction with Unblur or MotionCorr2 (5×5 patches) (*Grant and Grigorieff, 2015*; *Zheng et al., 2017*). The parameters of the contrast transfer function were estimated by ctffind4 or GCTF (*Rohou and Grigorieff, 2015*; *Zhang, 2016*). Following motion correction, ~5 k particles from a subset of the images were interactively selected using RELION to generate templates representing different views for automated particle selection with RELION autopicking (*Scheres, 2012*) or gautomatch (https://www.mrc-lmb.cam.ac.uk/kzhang/). The autopicked particles were manually inspected to remove false positives. The resulting particle images were then subjected to 2D classification in RELION to remove particles belonging to low-abundance classes and to generate projection averages for initial model generation with EMAN2 imposing C4 symmetry (*Tang et al., 2007*) or cryoSPARC (*Punjani et al., 2017*).

For the $Ca^{2+}$-bound (open) conformation of hsSlo1-β4 complex, ~483 k particle images were selected from 5410 micrographs following 2D classification in RELION (*Scheres, 2012*). Orientation and translational parameters for the ~483 k particle images were then refined with the auto-refine algorithm of RELION, using the EMAN2-generated initial model as a reference. The refined particle images were subjected to RELION's 3D classification algorithm without a mask, skipping image alignment. Orientation and translational parameters for the 133 k particle images in the best class were refined using the auto-refine algorithm of RELION, resulting in a map with a resolution of 3.8 Å before postprocessing. The refined particle images were subjected to another round of 3D classification without image alignment, resulting in one major class. Orientation and translational parameters for the 118 k particle images in this class were refined using the auto-refine algorithm of RELION. The rotational and translational parameters determined by RELION were used as the input for further refinement by FrealignX, during which the resolution of the reference map used for alignment was limited to 6 Å to minimize over-refinement, resulting in a final map that achieved a resolution of 3.2 Å as assessed by Fourier shell correlation using the 0.143 cut-off criterion (*Figure 1—figure supplements 2–3* and *Table 1*) (*Lyumkis et al., 2013*; *Grant et al., 2018*). The final map was sharpened using an isotropic b-factor of $-100$ $Å^2$ prior to model building and coordinate refinement. To improve the map of the extracellular domain of β4, focused 3D classification around the β4 subunit was performed on the two best classes from the first round of 3D classification. The 62 k particles from the best class were further refined using FrealignX masking around β4, resulting in a final map that achieved a resolution of 3.9 Å as assessed by Fourier shell correlation using the 0.143 cut-off criterion. This focus-refined map was used for initial de novo building of the β4 EC domain.

For the $Ca^{2+}$-free (closed) conformation of hsSlo1-β4 complex, ~269 k particle images were selected from 3405 micrographs following 2D classification in cryoSPARC (*Punjani et al., 2017*). Ab initio reconstruction of the 269 k particles (requesting three classes) resulted in two good classes with 198 k particles. Orientation and translational parameters for these particles were refined with the auto-refine algorithm of RELION (*Scheres, 2012*). The refined particle images were subjected to RELION's 3D classification algorithm without image alignment. One best class (~43 k particles) out of the requested six classes was refined using the auto-refine algorithm of RELION, resulting in a map that achieved a resolution of 4.2 Å before postprocessing. The rotational and translational parameters determined by RELION were used as the input for further refinement by FrealignX, during which the resolution of the reference map used for alignment was limited to 6 Å to minimize

over-refinement, resulting in a final map that achieved a resolution of 3.5 Å as assessed by Fourier shell correlation using the 0.143 cut-off criterion (*Figure 1—figure supplement 4* and *Table 1*) (*Lyumkis et al., 2013*; *Grant et al., 2018*). The map was sharpened using an isotropic b-factor of $-100$ Å$^2$ prior to model building and coordinate refinement.

For the Ca$^{2+}$-bound (open) conformation of hsSlo1, ~93 k particle images were selected from 1215 micrographs following 2D classification in RELION (*Scheres, 2012*). Orientation and translational parameters for these particle images were then refined with the auto-refine algorithm of RELION, using the EMAN2-generated initial model as a reference. The refined particle images were subjected to RELION's 3D classification algorithm without image alignment. One best class out of the requested four classes, accounting for ~30% of the total input, was refined using the auto-refine algorithm of RELION, resulting in a map that achieved a resolution of 4.5 Å before postprocessing. The rotational and translational parameters determined by RELION were used as the input for 40 additional cycles of refinement by FREALIGN, during which the resolution of the reference map used for alignment was limited to 6 Å to minimize over-refinement, resulting in a final map that achieved a resolution of 3.8 Å as assessed by Fourier shell correlation using the 0.143 cut-off criterion (*Figure 1—figure supplement 4* and *Table 1*) (*Lyumkis et al., 2013*; *Grant et al., 2018*). The map was sharpened using an isotropic b-factor of $-100$ Å$^2$ prior to model building and coordinate refinement.

For the Ca$^{2+}$-free (closed) conformation of hsSlo1, orientation and translational parameters of ~437 k autopicked particle images from 1292 micrographs were refined with the auto-refine algorithm of RELION using an initial model generated from cryoSPARC (*Scheres, 2012*; *Punjani et al., 2017*). The refined particle images were subjected to RELION's 3D classification algorithm without image alignment, requesting six classes. Orientation and translational parameters for the 54 k particle images in the best class was refined using the auto-refine algorithm of RELION, resulting in a map that achieved a resolution of 4.9 Å before postprocessing. The rotational and translational parameters determined by RELION were used as the input for 40 additional cycles of refinement by FREALIGN, during which the resolution of the reference map used for alignment was limited to 7 Å to minimize over-refinement, resulting in a final map that achieved a resolution of 4.0 Å as assessed by Fourier shell correlation using the 0.143 cut-off criterion (*Figure 1—figure supplement 4* and *Table 1*) (*Lyumkis et al., 2013*; *Grant et al., 2018*). The map was sharpened using an isotropic b-factor of $-200$ Å$^2$ prior to model building and coordinate refinement.

## Model building and refinement

The transmembrane domain from the cryo-EM structure of open aplysia Slo1 (PDB 5TJ6) and the X-ray crystal structure of the human Slo1 gating-ring (PDB 3MT5) were docked into the cryo-EM density map of the Ca$^{2+}$-bound Slo1-β4 complex using UCSF Chimera and then manually rebuilt in Coot to fit the density (*Pettersen et al., 2004*; *Emsley et al., 2010*). For the β4 subunit, the cryo-EM density map from a focus-classified and refined class was used for initial de novo building. The two transmembrane helices TM1 and TM2 were built by first placing secondary structure elements into the density. Once the backbone was traced, the sequence was registered by the assignment of large sidechains. The extracellular domain of β4 was built by identifying large sidechains, four pairs of disulfide bonds as well as the two N-glycosylation sites. An essentially complete model of β4 subunit was built with the exception of the very N-terminal six residues (aa 1–6) and C-terminal five residues (aa 206–210) for which no density was visible. The Slo1-β4 model after manual rebuilding in Coot was subjected to real-space refinement in Phenix (*Afonine et al., 2018*). The final model after a few iterations of real-space refinement and manual rebuilding has good geometry and contains amino acids 16–54, 91–569, 577–615, 681–833, and 871–1056 of the α subunit, and amino acids 7–205 of the β4 subunit (*Table 1*).

The atomic model of the Ca$^{2+}$-bound (open) Slo1-β4 complex was used as a starting model for the other three states followed by multiple rounds of manual rebuilding in Coot and real-space refinement with Phenix (*Emsley et al., 2010*; *Afonine et al., 2018*). The final models all have good geometry (*Table 1*). Figures were prepared using PyMOL (Molecular Graphics System, Version 2.2.0 Schrodinger, LLC) and Chimera (*Pettersen et al., 2004*).

## Mutagenesis

HsSlo1$_{EM}$, wild-type human β4 and β1 subunits were subcloned into a pGEM vector. Guided by the structure, in particular the relative location of the bound lipid molecules, we divided the entire β4 sequence into 10 regions: 1) the unresolved N-terminus ('N-tail', aa 1–6); 2) the short loop preceding TM1 ('N-loop', aa 7–12); 3) TM1 near the inner membrane leaflet headgroup layer ('TM1 ILH', aa 13–19); 4) TM1 near the inner membrane leaflet acyl chain layer ('TM1 ILA', aa 20–29); 5) TM1 near the outer membrane leaflet layer ('TM1 OL', aa 30–48); 6) EC domain (aa 49–163); 7) TM2 near the outer membrane leaflet layer ('TM2 OL', aa 164–180); 8) TM2 near the inner membrane leaflet acyl chain layer ('TM2 ILA', aa 181–190); 9) TM2 near the inner membrane leaflet headgroup layer ('TM2 ILH', aa 191–205); and 10) unresolved C-terminus ('C-tail', aa 206–210) (*Figure 4B*). Mutants were made by replacing one or more of these 10 regions of β4 mostly with the equivalent residues from the β1 subunit based on the sequence alignment (*Figure 2—figure supplement 1A* and *Table 2*). All the mutants were generated using PCR and incorporation of the mutation(s) was verified by sequencing (GeneWiz).

## Excised Inside-out patch recordings

The constructs used for cryo-EM sample preparation: hsSlo1$_{EM}$, wild-type human β4 and β1 in pEG BacMam vector, were used for measuring the Ca$^{2+}$ sensitivity of the channel in HEK293T cells (ATCC) in voltage-clamp inside-out patch configuration.

0.5 μg of hsSlo1$_{EM}$ alone or 0.5 μg of hsSlo1$_{EM}$ together with 0.5 μg of human β4 or β1 were transfected into HEK293T cells at about 50–60% confluency using FuGENE HD transfection reagent following manufacturer's instructions (Promega). Cells were transferred to 30℃ after transfection and recordings were carried out 18–24 hr post-transfection.

Pipettes of borosilicate glass (Sutter Instruments; BF150-86-10) were pulled to ~2–3 MΩ resistance with a micropipette puller (Sutter Instruments; P-97) and polished with a microforge (Narishige; MF-83). All recordings were performed at room temperature in voltage-clamp excised inside-out patch configuration with an Axopatch 200B amplifier (Molecular Devices), Digidata 1440A analogue-to-digital converter interfaced with a computer, and pClamp10.5 software (Axon Instruments, Inc) for controlling membrane voltage and data acquisition. The recorded signal was filtered at 1 kHz and sampled at 10 kHz.

The bath solution contained 20 mM HEPES-NaOH, 136 mM KGluconate, 4 mM KCl and 10 mM Glucose, pH 7.4 (adjusted with NaOH) with an osmolarity of ~300 Osm/L. The bath solution supplemented with 2 mM MgCl$_2$ was used as the pipette solution. Solutions used for the Ca$^{2+}$-titration experiments contained the bath solution supplemented with an increasing amount of CaCl$_2$: 0.5 μM, 2.5 μM, 10 μM, 50 μM. Note here the Ca$^{2+}$ concentrations refer to the amount of Ca$^{2+}$ added from a stock of CaCl$_2$, not the free [Ca$^{2+}$]. Ca$^{2+}$-titration was achieved with local perfusion using a fast-pressurized microperfusion system (ALA Scientific; ALAVC3 × 8 PP). At each Ca$^{2+}$ concentration, the ionic current was measured with a voltage-family protocol.

## Two-electrode voltage clamp (TEVC) recordings

HsSlo1$_{EM}$, wild-type human β1, wild-type human β4 and β4 mutants in a pGEM vector were used for expression in Xenopus oocytes. cRNAs were prepared from NdeI linearized plasmids using Ampli-Cap-Max T7 high yield message maker kit (CELLSCRIPT) and purified with mMESSAGE mMACHINE kit (ThermoFisher Scientific). cRNA concentration was estimated based on agarose gel.

Xenopus oocytes were harvested from mature female *Xenopus laevis* and defolliculated by collagenase treatment for 1–2 hr at room temperature. Oocytes were then rinsed thoroughly and stored in ND96 solution (96 mM NaCl, 2 mM KCl, 1.8 mM CaCl$_2$, 1.0 mM MgCl$_2$, 5 mM HEPES, 50 μg/ml gentamycin, pH 7.6 with NaOH). Defolliculated oocytes were selected 2–4 hr after collagenase treatment and injected with cRNA the next day. For co-expression of hsSlo1 and β, cRNAs at a ratio of 1: 2 (w:w) were injected. The injected oocytes were incubated in ND96 solution before recording. Recordings were usually carried out 1–2 days post-injection for ionic current measurements. All oocytes were stored in an incubator at 18℃.

All recordings were performed at room temperature in two-electrode voltage-clamp configuration with an oocyte clamp amplifier (OC-725C, Warner Instrument Corp.), Digidata 1550B analogue-to-digital converter interfaced with a computer, and pClamp11.0.1 software (Axon Instruments, Inc)

for controlling membrane voltage and data acquisition. The recorded signal was filtered at 1 kHz and sampled at 10 kHz.

Oocytes were recorded with bath solution of either low $Ca^{2+}$ ND96 recording solution (96 mM NaCl, 2 mM KCl, 0.3 mM $CaCl_2$, 1.0 mM $MgCl_2$, 5 mM HEPES, pH 7.6 with NaOH) or high K recording solution (98 mM KCl, 0.3 mM $CaCl_2$, 1.0 mM $MgCl_2$, 5 mM HEPES, pH 7.6 with NaOH). To investigate voltage-dependent channel activation and deactivation, oocytes were held at either −80 mV for recordings in low $Ca^{2+}$ ND96 solution or 0 mV for recordings in high K solution, with pulse potential starting from the holding potential and ending between +120 mV and +140 mV in 10 mV increments. The repolarization potential was +40 mV.

## Data analysis

For voltage-dependent channel activation recordings at a series of $Ca^{2+}$ concentrations in excised inside-out patch experiments, the amount of current at the repolarization step (i.e. tail current), typically measured 4–5 ms after the depolarization step when most of the capacitive current had decayed, was normalized against the maximal current ($I/I_{max}$) at that concentration and plotted as a function of the depolarization voltage (I-V plot). This voltage-dependent activation plot was fitted with the two-state Boltzmann function:

$$\frac{I}{I_{max}} = \frac{1}{1 + e^{-\left(\frac{ZF}{RT}(V - V_m)\right)}}$$

where $I/I_{max}$ is the fraction of the maximal current, V is the depolarization voltage to open the channels, $V_m$ is the voltage at which the channels have reached 50% of their maximal current, F is the Faraday constant, R is the gas constant, T is the absolute temperature, and Z is the apparent valence of voltage dependence.

For the voltage-dependent channel activation recordings made in oocytes with TEVC, the channel activation time course at the depolarization step (100 mV) and the deactivation time course at the repolarization step (40 mV) (typically spanning from 4 to 5 ms after the depolarization step or repolarization step when most of the capacitive current had decayed until the current has reached its steady state) were quantified by fitting with a single exponential function:

$$I(t) = A\left(1 - e^{\frac{-t}{\tau}}\right) + C$$

All data analysis and fits were carried out with the Clampfit software (Axon Instruments, Inc).

## Acknowledgements

We thank Z Yu and R Huang at the Howard Hughes Medical Institute Janelia Cryo-EM facility for assistance in data collection of the open Slo1-β4 complex structure; M Ebrahim and J Sotiris at the Evelyn Gruss Lipper Cryo-EM Resource Center at Rockefeller University for assistance in data collection of the closed Slo1-β4 complex and open/closed α alone Slo1 structures; thank H Funabiki's Laboratory (Rockefeller University) for generously providing us frogs, J Sun for help with preparing oocytes; RK Hite (MSKCC) and N Paknejad (MSKCC) for assistance with data processing; EC Brown for comments on the manuscript; and members of the MacKinnon lab and Chen lab (Rockefeller University) for assistance. This work was supported in part by GM43949. RM is an investigator of the Howard Hughes Medical Institute.

## Additional information

### Funding

| Funder | Grant reference number | Author |
|---|---|---|
| National Institutes of Health | GM43949 | Roderick MacKinnon |
| Howard Hughes Medical Institute | | Roderick MacKinnon |

The funders had no role in study design, data collection and interpretation, or the decision to submit the work for publication.

## Author contributions
Xiao Tao, Conceptualization, Data curation, Formal analysis, Validation, Methodology, Project administration; Roderick MacKinnon, Conceptualization, Resources, Formal analysis, Supervision, Funding acquisition, Validation, Investigation

## Author ORCIDs
Xiao Tao (iD) https://orcid.org/0000-0002-9381-7903
Roderick MacKinnon (iD) https://orcid.org/0000-0001-7605-4679

## Decision letter and Author response
Decision letter https://doi.org/10.7554/eLife.51409.sa1

# Additional files

## Supplementary files
• Transparent reporting form

## Data availability
The B-factor sharpened 3D cryo-EM density maps and atomic coordinates of the Ca2+-bound (open) hsSlo1-beta4 complex (accession number EMD-21025 and 6V22), the Ca2+-free (closed) hsSlo1-beta4 complex (accession number EMD-21028 and 6V35), the Ca2+-bound (open) hsSlo1 (accession number EMD-21029 and 6V38), and the Ca2+-free (closed) hsSlo1 (accession number EMD-21036 and 6V3G) have been deposited in the Worldwide Protein Data Bank (wwPDB).

The following datasets were generated:

| Author(s) | Year | Dataset title | Dataset URL | Database and Identifier |
|---|---|---|---|---|
| Tao X, MacKinnon R | 2019 | Single particle cryo-EM structure of Ca2+-bound (open) hsSlo1-beta4 complex | https://www.rcsb.org/structure/6V22 | Protein Data Bank, PDB 6V22 |
| Tao X, MacKinnon R | 2019 | Single particle cryo-EM structure of Ca2+-free (closed) hsSlo1-beta4 complex | https://www.rcsb.org/structure/6V35 | Protein Data Bank, PDB 6V35 |
| Tao X, MacKinnon R | 2019 | Single particle cryo-EM structure of Ca2+-bound (open) hsSlo1 | https://www.rcsb.org/structure/6V38 | Protein Data Bank, PDB 6V38 |
| Tao X, MacKinnon R | 2019 | Single particle cryo-EM structure of Ca2+-free (closed) hsSlo1 | https://www.rcsb.org/structure/6V3G | Protein Data Bank, PDB 6V3G |
| Tao X, MacKinnon R | 2019 | Single particle cryo-EM structure of Ca2+-bound (open) hsSlo1-beta4 complex | http://www.ebi.ac.uk/pdbe/entry/emdb/EMD-21025 | EMDataBank, EMD-21025 |
| Tao X, MacKinnon R | 2019 | Single particle cryo-EM structure of Ca2+-free (closed) hsSlo1-beta4 complex | https://www.ebi.ac.uk/pdbe/entry/emdb/EMD-21028 | EMDataBank, EMD-21028 |
| Tao X, MacKinnon R | 2019 | Single particle cryo-EM structure of Ca2+-bound (open) hsSlo1 | https://www.ebi.ac.uk/pdbe/entry/emdb/EMD-21029 | EMDataBank, EMD-21029 |
| Tao X, MacKinnon R | 2019 | Single particle cryo-EM structure of Ca2+-free (closed) hsSlo1 | https://www.ebi.ac.uk/pdbe/entry/emdb/EMD-21036 | EMDataBank, EMD-21036 |

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
