## [Decision Letter]

**Acceptance summary:**

Slo or BK calcium and voltage activated potassium channels serve a wide variety of roles in diverse tissues where their properties are modified in tissue-specific ways by alternative splicing, association of different accessory (β and γ) subunits and protein modification. Tao and MacKinnon present a CryoEM derived structure of the human Slo chanel containing the brain associated β4 subunit, an auxiliary subunit that alters channel gating characteristics. They present atomic models of four different states of the channel, in the presence and absence of Ca^2+^ and presence and absence of the β4 subunit. Their study confirms that the human channel structure does not substantially differ from the previously solved *Aplysia* channel and maintains its general structure when bound to the β4 subunit.

**Decision letter after peer review:**

[Editors’ note: minor issues and corrections have not been included, so there is not an accompanying Author response.]

Thank you for submitting your article "Molecular structures of the human Slo1 K + channel in complex with b4" for consideration by *eLife*. Your article has been reviewed by three peer reviewers, and the evaluation has been overseen by a Reviewing Editor and Kenton Swartz as the Senior Editor. The following individuals involved in review of your submission have agreed to reveal their identity: Niko Grigorieff (Reviewer #2).

The reviewers have discussed the reviews with one another and the Reviewing Editor has drafted this decision to help you prepare a revised submission.

Summary:

The paper describes the first structures of Slo1 (BK) potassium channels in complex with a regulatory β subunit and as such makes significant and important contribution to the understanding of these important ion channels.

Essential revisions:

The reviewers find the work to be of high quality, but they bring up several fairly minor issues that need to be addressed. These are clearly described in the individual reviews enclosed below, and should be easily addressed.

Reviewer #1:

This is a relatively straightforward structural paper providing the first structural information regarding a BK regulatory subunit in association with full-length BK channels. The paper includes new structures for both relatively full-length human Slo1 in both putative open and closed conformations, with and without the β4subunit. The structural work is complemented by functional evaluation of which part of β4 may impact on channel function, taking advantage of the chimeras designed around segments based on the β structures. Although not major insights are gained from the functional work, it does highlight a few loci that may motivate future investigation.

Reviewer #2:

Tao and MacKinnon have determined high-resolution cryo-EM structures of human Slo1 ion channel, a channel that is gated by voltage and Ca^2+^. They show atomic models of four different states of the channel, corresponding to combinations of presence and absence of Ca^2+^ and presence and absence of the β4 subunit, an auxiliary subunit that alters channel gating characteristics. The structures of human Slo1 are very similar to homologous *Aplysia* Slo1, which the authors have studied previously, displaying almost identical Ca^2+^ dependent conformational changes. However, the structure of any of the tissue-specific β subunits was hitherto unknown, and the structure shown in the present manuscript represents one of the main results. The authors provide detailed descriptions of the interactions between β4 and the channel, forming an octameric complex with 1:1 stoichiometry.

Since the structures of the channel in the presence and absence of β4 are virtually identical, it is not obvious how β4 affects channel gating. β4 has two transmembrane segments, and based on the new cryo-EM structures, the authors suggest that binding to the channel may alter its interaction with lipid molecules in the native membrane environment, potentially affecting gating behavior. Furthermore, contacts between β4 and the channel may shift its equilibrium between open and closed states, which would not alter the conformations of the fully open and fully closed states observed under the conditions used for the cryo-EM experiments. The β4-bound structures also explain how the auxiliary subunit interferes with toxin binding, which was previously observed for Slo1.

The cryo-EM work follows standard procedures and is of high quality, permitting reliable building and refinement of atomic models. The main weakness of this manuscript is the lack of a clear molecular mechanism of how β4 alters the characteristics of the Slo1 channel. The authors perform mutagenesis of β4, as well as electrophysiology, highlighting the β4 N-terminal tail as the segment that conveys the biggest change. Unfortunately, this part of β4 is disordered and not visible in any of the reconstructions, not clarifying the mechanism. The structures also do not offer any insight into how β4 affects the voltage sensor. The main advance of this study is therefore the structure of β4, in complex with the Slo1 α subunit. As explained by the authors, this will serve as a basis for further studies to design experiments and test hypotheses on the molecular mechanism of β subunits binding to Slo channels. Given that there was no structure of a β subunit known, and that the present structures show that some previous conclusions based on more indirect experiments were wrong, this work is of sufficiently broad interest and significance for publication in *eLife*.

Reviewer #3:

The paper by Tao and MacKinnon provides us, for the first time, with the structure of the BK channel formed by the αβ4 complex. The structure of the BK channel, obtained at a resolution of 3.2 Å, shows a BK channel in which four β4 subunits are intercalated in between two α-subunit voltage sensors. The β4 does not introduce large conformational changes when BK structures with and without the β4 subunit or with and without Ca^2+^ are compared. The authors hypothesize that "β4 has the role of modulating the relative stability of pre-existing conformations". The paper provides a wealth of information about the detailed structure of the β4 in the αβ4 complex and the molecular interactions between the α and the β4 subunit. In this regard, it is important to mention that TM2 forms an interface with lipid molecules and that on the extracellular side, β4 forms a well-defined tetrameric domain, defining a crown-like structure. The later structure gives a natural explanation to the observation that the rates of association and dissociation of scorpion toxins are highly modified in αβ4 BK channels. The β4 structure and electrophysiological and mutagenesis experiments also indicate that the interaction between the N-terminus of TM1 of β4 and some domains of the gating ring is involved in the modulation of channel gating.

This is an important paper that provides us with a structure that without a doubt, will allow a further understanding of how β subunits modulate BK channel gating.